# RAPSYN-mediated neddylation of BCR-ABL alternatively determines the fate of Philadelphia chromosome-positive leukemia

Mengya Zhao[1†], Beiying Dai[2†], Xiaodong Li[1†], Yixin Zhang[2†], Chun Qiao[2,3], Yaru Qin[2], Zhao Li[1], Qingmei Li[1], Shuzhen Wang[1]*, Yong Yang[2]*, Yijun Chen[1,2,4]*

[1]Laboratory of Chemical Biology, School of Life Science and Technology, China Pharmaceutical University, Nanjing, China; [2]State Key Laboratory of Natural Medicines, China Pharmaceutical University, Nanjing, China; [3]Department of Hematology, The First Affiliated Hospital of Nanjing Medical University, Jiangsu Province Hospital, Nanjing, China; [4]Chongqing Innovation Institute of China Pharmaceutical University, Chongqing, China

*For correspondence:
shuzhenwang@cpu.edu.cn (SW);
yy@cpu.edu.cn (YY);
yjchen@cpu.edu.cn (YC)

[†]These authors contributed equally to this work

**Abstract** Philadelphia chromosome-positive (Ph[+]) leukemia is a fatal hematological malignancy. Although standard treatments with tyrosine kinase inhibitors (TKIs) have achieved remarkable success in prolonging patient survival, intolerance, relapse, and TKI resistance remain serious issues for patients with Ph[+] leukemia. Here, we report a new leukemogenic process in which RAPSYN and BCR-ABL co-occur in Ph[+] leukemia, and RAPSYN mediates the neddylation of BCR-ABL. Consequently, neddylated BCR-ABL enhances the stability by competing its c-CBL-mediated degradation. Furthermore, SRC phosphorylates RAPSYN to activate its NEDD8 E3 ligase activity, promoting BCR-ABL stabilization and disease progression. Moreover, in contrast to in vivo ineffectiveness of PROTAC-based degraders, depletion of RAPSYN expression, or its ligase activity decreased BCR-ABL stability and, in turn, inhibited tumor formation and growth. Collectively, these findings represent an alternative to tyrosine kinase activity for the oncoprotein and leukemogenic cells and generate a rationale of targeting RAPSYN-mediated BCR-ABL neddylation for the treatment of Ph[+] leukemia.

## eLife assessment

In this **important** study, the authors describe a novel function for RAPSYN in bcr-abl fusion associated leukemia, presenting **convincing** evidence that RAPSYN stabilizes the oncogenic BCR-ABL fusion protein. Compared to an earlier version of the manuscript, the authors have added data using primary samples that strengthen the conclusions.

## Introduction

Philadelphia chromosome-positive (Ph[+]) leukemia is a myeloproliferative neoplasm characterized by the reciprocal translocation between the long arms of chromosomes 9 and 22, t (9;22) (q34.1; q11.2) (*de Klein et al., 1982*; *Deininger et al., 2000*). This cytogenetic abnormality results in a *BCR-ABL* fusion gene, which encodes the chimeric protein BCR-ABL with enhanced tyrosine kinase activity (*Cortes et al., 2021*). Based on its oncogenic role in Ph[+] leukemia, BCR-ABL has been regarded as the most pivotal target for Ph[+] leukemia therapy, especially for chronic myeloid leukemia (CML). Tyrosine

**eLife digest** Chronic myeloid leukemia (CML for short) accounts for about 15% of all blood cancers diagnosed in adults in the United States. The condition is characterized by the overproduction of immature immune cells that interfere with proper blood function. It is linked to a gene recombination (a type of mutation) that leads to white blood cells producing an abnormal 'BCR-ABL' enzyme which is always switched on. In turn, this overactive protein causes the cells to live longer and divide uncontrollably.

Some of the most effective drugs available to control the disease today work by blocking the activity of BCR-ABL. Yet certain patients can become resistant to these treatments over time, causing them to relapse. Other approaches are therefore needed to manage this disease; in particular, a promising avenue of research consists in exploring whether it is possible to reduce the amount of the enzyme present in diseased cells.

As part of this effort, Zhao, Dai, Li, Zhang et al. focused on RAPSYN, a scaffolding protein previously unknown in CML cells. In other tissues, it has recently been shown to participate in neddylation – a process by which proteins receive certain chemical 'tags' that change the way they behave. The experiments revealed that, compared to healthy volunteers, RAPSYN was present at much higher levels in the white blood cells of CML patients. Experimentally lowering the amount of RAPSYN in CML cells led these to divide less quickly – both in a dish and when injected in mice, while also being linked to decreased levels of BCR-ABL.

Additional biochemical experiments indicated that RAPSYN sticks with BCR-ABL to add chemical 'tags' that protect the abnormal protein against degradation, therefore increasing its overall levels.

Finally, the team showed that SRC, an enzyme often dysregulated in emerging cancers, can activate RAPSYN's ability to conduct neddylation; such mechanism could promote BCR-ABL stabilization and, in turn, disease progression.

Taken together, these experiments indicate a new way by which BCR-ABL levels are controlled. Future studies should investigate whether RAPSYN also stabilizes BCR-ABL in patients whose leukemias have become resistant to existing drugs. Eventually, RAPSYN may offer a new target for overcoming drug-resistance in CML patients.

kinase inhibitors (TKIs) have been the main treatment option for Ph[+] leukemia, remarkably prolonging the patients' lifespan and improving their quality of life (*Hochhaus et al., 2020*; *Jabbour and Kantarjian, 2020*). However, most patients develop TKI resistance and relapse after long-term treatment (*Braun et al., 2020*). It is worth noting that the increase of BCR-ABL expression can affect the sensitivity to TKIs and eventually determine the rate of TKI resistance in patients with Ph[+] leukemia in addition to the mutations in the kinase domain (*Jabbour et al., 2007*). Mutations in the kinase domain can change the conformation of BCR-ABL, thus interfering with the binding between TKIs and BCR-ABL and resulting in decreased therapeutic efficacy (*Lussana et al., 2018*). In parallel, the increase of BCR-ABL expression can affect the sensitivity to TKIs and eventually determine the rate of disease progression and TKI resistance in patients with Ph[+] leukemia (*Barnes et al., 2005*). Therefore, effective degradation of BCR-ABL could address the issues on TKI resistance and leukemia-initiating cells (LICs), and PROTAC-based protein degradation strategy may represent a new therapeutic approach (*Békés et al., 2022*). Currently, based on different ubiquitin E3 ligases, including VHL, CRBN, and IAP, PROTAC-based degraders at nM level have shown significant degradation of BCR-ABL in CML cell lines, cell lines carrying mutations in BCR-ABL as well as patient-derived primary cells containing multiple BCR-ABL mutations (*Demizu et al., 2016*; *Lai et al., 2016*; *Shimokawa et al., 2017*; *Zhao et al., 2019*; *Burslem et al., 2019*; *Liu et al., 2022*). Unfortunately, the excellent cellular activity by the PROTAC-based degraders has not been able to translate to in vivo efficacy, even in rare examples of xenografted mouse models (*Zhao et al., 2019*; *Jiang et al., 2021b*), resulting in uncertain usefulness of these degraders. Nonetheless, the unsatisfactory results are not really surprising because the underlying mechanism of elevated BCR-ABL expression remains largely unclear.

Receptor-associated protein of the synapse (RAPSYN) has been identified as a classic synaptic adaptor protein that binds to the acetylcholine receptor (AChR) and several cytoskeleton-associated proteins, contributing to AChR clustering and neuromuscular junction formation (*Huh and Fuhrer,*

2002; *Witzemann et al., 2013*). Later, RAPSYN was found to exert NEDD8 E3 ligase activity to catalyze the neddylation for AChR aggregation (*Li et al., 2016*). Despite the extensive studies of RAPSYN in muscular and neuronal cells and tissues (*Legay and Mei, 2017*; *Li et al., 2018*), with regard to its involvement in leukemogenesis, there is no available information thus far except for our previous finding. Previously, RAPSYN was found to be located in the cytosol of the typical Ph$^+$ leukemia cell line K562 when a small molecule was used to probe its binding proteins using a proteomics approach (*Wang et al., 2015*). Because of its newly identified E3 ligase activity for neddylation and its occurrence in the Ph$^+$ leukemia cell line, we speculated that RAPSYN might contribute to Ph$^+$ leukemia development through its enzymatic activity instead of only serving as a scaffolding protein.

As a type of post-translational modification (PTM), neddylation is sequentially catalyzed by the neuronal precursor cell-expressed developmentally downregulated protein 8 (NEDD8)-activating enzyme E1 (NAE1), NEDD8-conjugating enzyme E2 (UBE2M/UBC12 or UBE2F), and a substrate-specific NEDD8 E3 ligase to complete the covalent conjugation of NEDD8 to a lysine residue of its substrates (*van der Veen and Ploegh, 2012*; *Enchev et al., 2015*). The neddylation of proteins can be reversed by deneddylases such as NEDP1. In the last two decades, accumulating evidence indicated the strong involvement of dysregulated neddylation in tumor progression, neurodegenerative and cardiac diseases, aberrant immunoregulation and others (*Ying et al., 2018*; *Zhou et al., 2019*; *Li et al., 2020*; *Yao et al., 2020*), which rationalizes the modulation of neddylation as a feasible therapeutic strategy.

In this study, we report that RAPSYN is highly expressed along with BCR-ABL in patients with Ph$^+$ leukemia and promotes disease progression, presumably by stabilizing the BCR-ABL fusion protein *via* neddylation. The neddylation of BCR-ABL by RAPSYN subsequently competes its ubiquitination-dependent degradation to increase the stability of BCR-ABL. Additionally, the NEDD8 E3 ligase activity of RAPSYN can be substantially increased by SRC-mediated phosphorylation, leading to enhanced stability and activity of RAPSYN.

## Results

### High protein levels of RAPSYN promoted Ph$^+$ leukemia progression

Prior to investigating the biological roles of RAPSYN in the pathogenesis of Ph$^+$ leukemia, its expression at both mRNA and protein levels was analyzed. We analyzed mRNA levels of *RAPSN* in RNA-seq datasets of GSE13204, GSE13159, GSE138883, and GSE140385, and no difference of *RAPSN* mRNA levels in peripheral blood mononuclear cells (PBMCs) was found between CML patients and healthy donors (*Figure 1—figure supplement 1A*). Neither a publicly available database nor our collection of patient samples and cell lines showed a significant increase in mRNA levels (*Figure 1—figure supplement 1B, C*). The protein levels of RAPSYN were substantially elevated in the PBMCs of Ph$^+$ CML (#8–11) and the bone marrow of ALL (#7) patient samples in comparison to that of healthy donors (#1–6), which was in a direct accordance with the expression of BCR-ABL (*Figure 1A*). This co-expression of RAPSYN and BCR-ABL was also found in Ph$^+$ cell lines (*Figure 1B*), suggesting that the function of RAPSYN in Ph$^+$ leukemia could be closely related to BCR-ABL.

To examine the relationship between RAPSYN and Ph$^+$ leukemia progression, we first performed knockdown of its expression by using shRNAs. Whereas notable cytotoxicity following marked reduction of RAPSYN was observed in all tested Ph$^+$ leukemia cell lines and CML patient PBMCs (*Figure 1C, D, Figure 1—figure supplement 1D–F*), transduction with the shRNA for *RAPSN* did not affect cell viability of RAPSYN- and BCR-ABL-negative HS-5 cells, indicating the dependence on the presence and expression level of BCR-ABL. Conversely, exogenous expression of *RAPSN* rescued Ph$^+$ leukemia cells from shRNA-generated toxicity (*Figure 1E*). Knockdown of *RAPSN* also changed the phenotypes of Ph$^+$ leukemia cells, including proliferation, G0/G1 cell cycle arrest, and apoptosis (*Figure 1—figure supplement 1G–I*).

Next, we subcutaneously implanted sh*RAPSN* #3- or the empty vector-transduced K562 cells into NCG mice to establish a cell line-derived xenotransplantation mouse model (*Figure 1F*). Tumor growth was significantly inhibited by *RAPSN* silencing (*Figure 1G, H, Figure 1—figure supplement 1J*). Meanwhile, immunoblotting of tumor samples showed a notable downregulation of RAPSYN expression, along with the reduction of BCR-ABL levels (*Figure 1I*). After knockout of *RAPSN* for remarkable depletion of BCR-ABL expression in K562 cells (*Figure 1J, Figure 1—figure supplement*

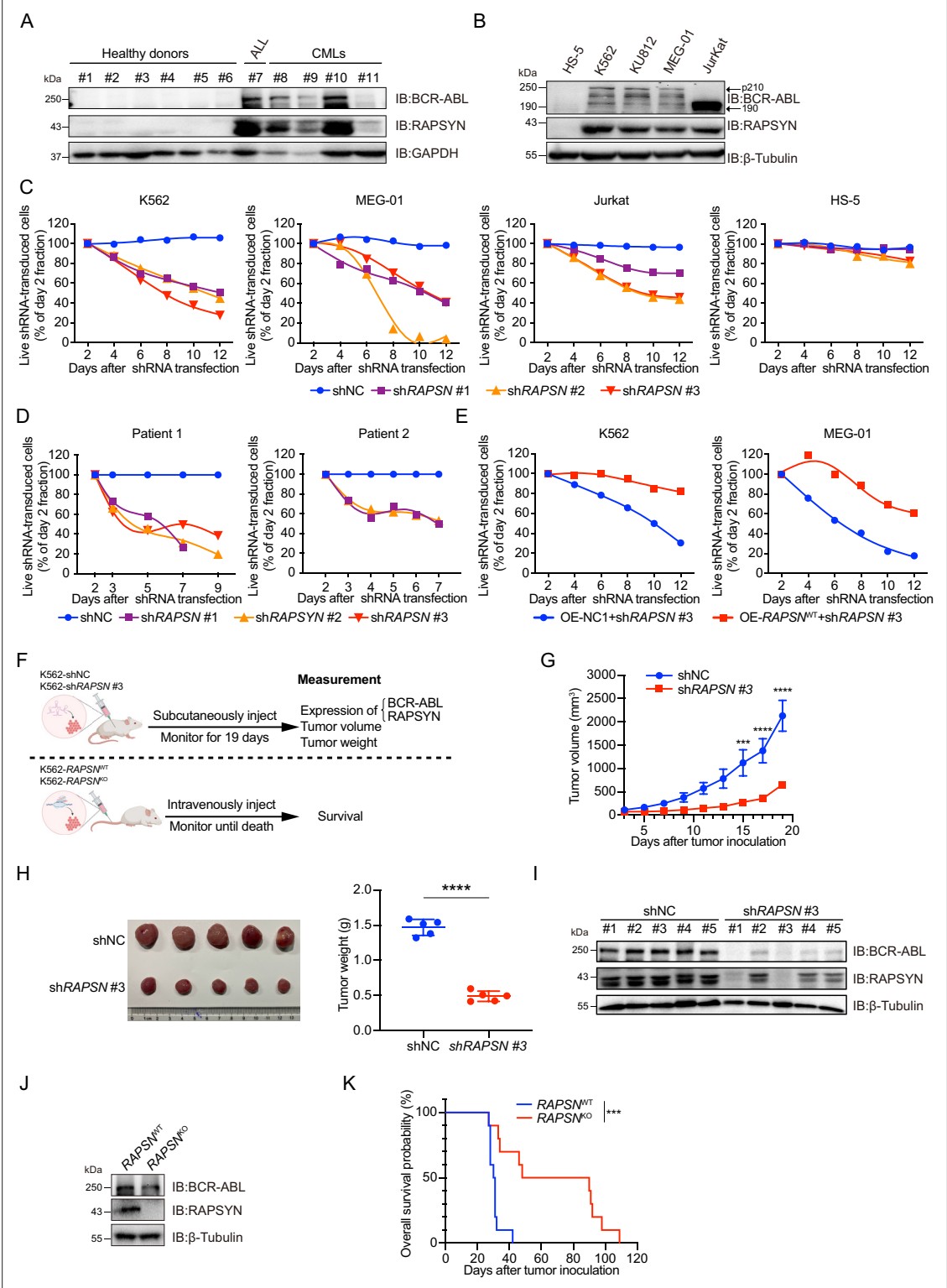

**Figure 1.** High protein levels of RAPSYN promotes Ph⁺ leukemia progression. (**A**) Immunoblots of RAPSYN and BCR-ABL in the peripheral blood mononuclear cells (PBMCs) of clinical samples. (**B**) Immunoblots of RAPSYN and BCR-ABL in Ph⁺ leukemic cells and normal bone marrow stromal cells (HS-5). (**C**) Cytotoxicity induced by shRNA-mediated *RAPSN* knockdown in leukemic and HS-5 cells. (**D**) Cytotoxicity induced by shRNA-mediated *RAPSN* knockdown in the PBMCs of chronic myeloid leukemia (CML) patients. (**E**) Rescue of leukemic cells from sh*RAPSN* #3-induced toxicity by exogenous expression of *RAPSN* cDNA or NC1. (**F**) An in vivo experimental design for testing the effects of RAPSYN on tumor growth and survival. (**G**) The growth curve of subcutaneous xenograft tumors was measured every 2 days from the third day after tumor inoculation for 19 days (five mice

*Figure 1 continued on next page*

*Figure 1 continued*

in each group). (**H**) Photograph and weight quantification of excised tumor xenografts from (**I**). (**I**) Immunoblots of RAPSYN and BCR-ABL in mouse xenograft tumor biopsies from K562 cells transduced with sh*RAPSN* #3 or shNC. (**J**) Immunoblots of RAPSYN and BCR-ABL in K562-*RAPSN*^WT and K562-*RAPSN*^KO cells. (**K**) Kaplan–Meier survival curve of NCG mice following intravenous injection of K562-*RAPSN*^WT or K562-*RAPSN*^KO cells, as shown in (**F**) (10 mice in each group). All data represent mean ± standard deviation (SD) of at least three independent experiments. p values were calculated using unpaired Student's *t*-test (G and H) or log-rank test (K). ***p < 0.001, ****p < 0.0001.

The online version of this article includes the following source data and figure supplement(s) for figure 1:

**Source data 1.** Original file for the Western blot analysis in *Figure 1A* (anti-BCR-ABL, anti-RAPSYN, anti-GAPDH).

**Source data 2.** PDF containing *Figure 1A* and original scan of the relevant Western blot analysis (anti-BCR-ABL, anti-RAPSYN, anti-GAPDH) with highlighted bands and sample labels.

**Source data 3.** Original file for the Western blot analysis in *Figure 1B* (anti-BCR-ABL, anti-RAPSYN, anti-β-Tubulin).

**Source data 4.** PDF containing *Figure 1B* and original scan of the relevant Western blot analysis (anti-BCR-ABL, anti-RAPSYN, anti-β-Tubulin) with highlighted bands and sample labels.

**Source data 5.** Original file for the Western blot analysis in *Figure 1I* (anti-BCR-ABL, anti-RAPSYN, anti-β-Tubulin).

**Source data 6.** PDF containing *Figure 1I* and original scan of the relevant Western blot analysis (anti-BCR-ABL, anti-RAPSYN, anti-β-Tubulin) with highlighted bands and sample labels.

**Source data 7.** Original file for the Western blot analysis in *Figure 1J* (anti-BCR-ABL, anti-RAPSYN, anti-β-Tubulin).

**Source data 8.** PDF containing *Figure 1J* and original scan of the relevant Western blot analysis (anti-BCR-ABL, anti-RAPSYN, anti-β-Tubulin) with highlighted bands and sample labels.

**Figure supplement 1.** The mRNA levels of *RAPSN* are not changed by Ph$^+$ leukemia, whereas inhibition of RAPSYN suppresses Ph$^+$ leukemia progression.

**Figure supplement 1—source data 1.** Original file for the Western blot analysis in *Figure 1—figure supplement 1E* (anti-RAPSYN, anti-GAPDH).

**Figure supplement 1—source data 2.** PDF containing *Figure 1—figure supplement 1E* and original scan of the relevant Western blot analysis (anti-RAPSYN, anti-GAPDH) with highlighted bands and sample labels.

*1L*), these cells along with the empty vector-transduced K562 cells were intravenously injected into NCG mice to establish the leukemogenic mouse model (*Figure 1F*). Consequently, the survival of tumor-bearing mice was profoundly prolonged by the knockout of *RAPSN* compared to the controls (*Figure 1K*). Altogether, our findings indicate that RAPSYN is highly expressed at protein level with the accordance to BCR-ABL in Ph$^+$ leukemia and its depletion results in inhibiting the progression of Ph$^+$ leukemia.

## RAPSYN directly neddylated BCR-ABL

Previous reports determined that both nicotinic AChR subunit $\alpha_7$ and muscarinic AChR subtypes M$_2$, M$_3$, and M$_4$ were involved in the cell proliferation of K562 cells (*Cabadak et al., 2011*; *Önder Narin et al., 2021*). Furthermore, RAPSYN was found to exert its NEDD8 E3 ligase activity toward AChR in neuronal systems (*Li et al., 2016*). To determine whether RAPSYN functioned in a similar manner in leukemogenic cells, we investigated whether RAPSYN promoted Ph$^+$ leukemia progression through neddylation of AChRs. Despite the expression of AChR subunits α7, M2, M3, and M4 at protein level in all tested Ph$^+$ leukemia cells, no change in their neddylation was observed upon RAPSYN ablation (*Figure 2—figure supplement 1A, B*). In addition, we also examined mRNA levels of RAPSYN-related neddylation enzymes, including E1 (NAE1), E2 (UBE2M), NEDD8, and NEDP1, in above GSE databases, and no significant differences of these neddylation-related genes were found between CML patients and healthy donors as well (*Figure 2—figure supplement 1C*). On the basis of the co-expression of RAPSYN and BCR-ABL, we postulated that RAPSYN could specifically mediate neddylation of BCR-ABL to promote Ph$^+$ leukemia development.

To test this hypothesis, reciprocal immunoprecipitation was performed to reveal a strong interaction between RAPSYN and BCR-ABL in Ph$^+$ leukemia cells (*Figure 2A*, *Figure 2—figure supplement 1D*). Similar results were obtained with exogenous expression in HEK293T cells (*Figure 2B*), further confirming the specific interaction of RAPSYN with BCR-ABL. Furthermore, GST pull-down assay with purified proteins displayed specific binding of GST-tagged RAPSYN to His-tagged BCR-ABL (*Figure 2C*), indicating that BCR-ABL is the primary target of RAPSYN-mediated neddylation. Domain mapping revealed that the Δ1 domain (1–927 aa) of BCR-ABL was responsible for the interaction with RAPSYN (*Figure 2D*).

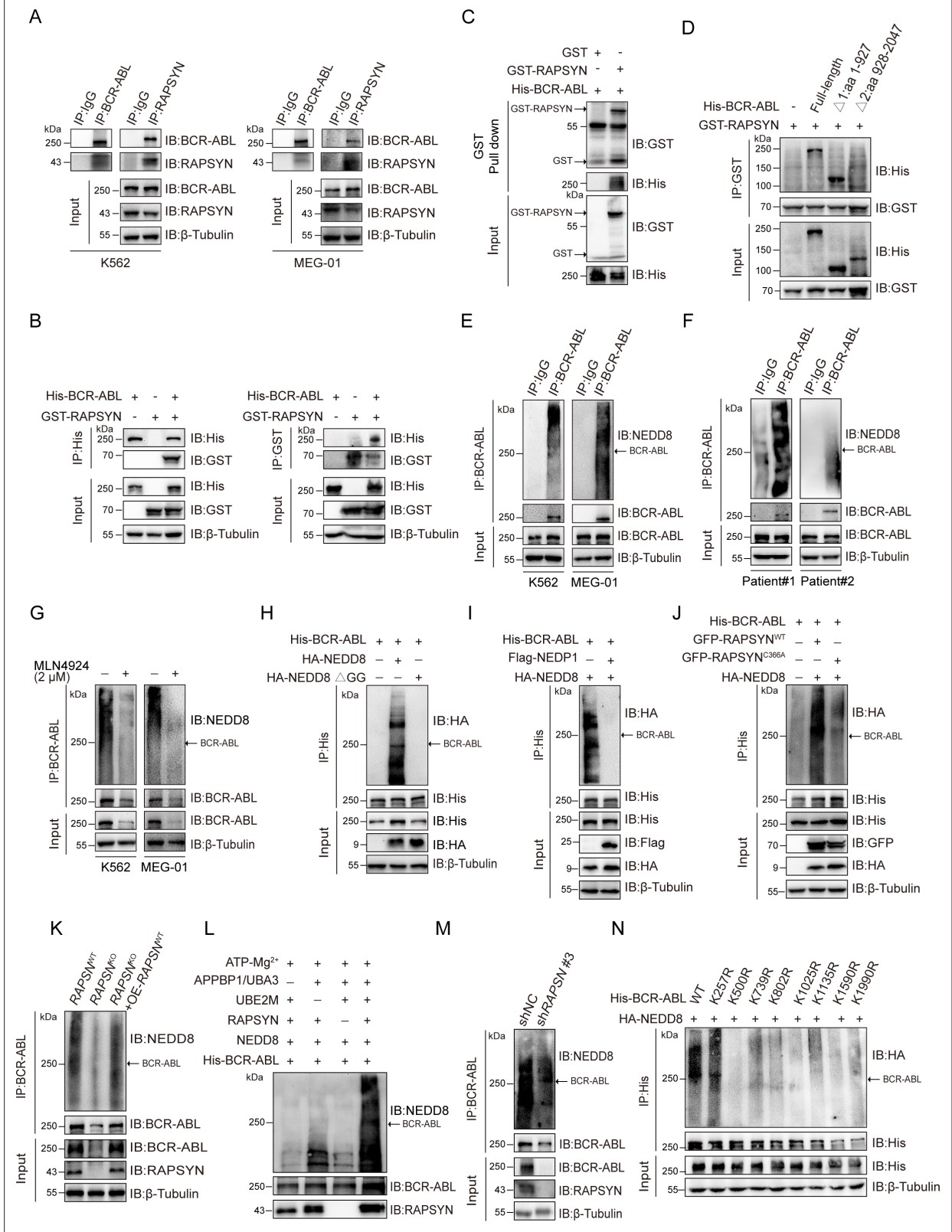

**Figure 2.** RAPSYN neddylates BCR-ABL. (**A**) Co-immunoprecipitation of BCR-ABL and RAPSYN in leukemic cells. (**B**) Immunoblots of GST and His after immunoprecipitation of His or GST in HEK293T cells transfected with His-tagged BCR-ABL and GST-tagged RAPSYN. (**C**) Immunoblots of GST and His following GST pull-down after in vitro incubation of purified His-tagged BCR-ABL and GST or GST-tagged RAPSYN. (**D**) His-immunoblots of GST immunoprecipitates from HEK293T cells transfected with GST-tagged RAPSYN alone or in combination with His-tagged full-length or truncated BCR-

*Figure 2 continued on next page*

*Figure 2 continued*

ABL (Δ1: aa 1–927, Δ2: aa 928–2047). (**E**) Analysis of BCR-ABL neddylation levels in leukemic cells. (**F**) Analysis of BCR-ABL neddylation levels in primary chronic myeloid leukemia (CML) peripheral blood mononuclear cells (PBMCs). (**G**) Analysis of BCR-ABL neddylation levels in leukemic cells treated with MLN4924 or dimethyl sulfoxide (DMSO) for 24 hr. (**H**) HA-immunoblots of His-immunoprecipitate from HEK293T cells transfected with His-tagged BCR-ABL and HA-tagged NEDD8 or NEDD8 ΔGG. (**I**) HA-immunoblots of His-immunoprecipitate from HEK293T cells transfected with indicated constructs. (**J**) HA-immunoblots after immunoprecipitation of His-antibody in HEK293T cells transfected with His-tagged BCR-ABL, HA-tagged NEDD8, GFP-tagged WT RAPSYN, or RAPSYN-C366A. (**K**) Analysis of BCR-ABL neddylation levels in K562 WT, *RAPSN* KO, and *RAPSN* KO with exogenous expression of a *RAPSN* cDNA cells. (**L**) Assessment of BCR-ABL neddylation by RAPSYN in vitro. Recombinantly expressed and purified RAPSYN and BCR-ABL were incubated with APPBP1/UBA3, UBE2M, or NEDD8 for in vitro neddylation assay. (**M**) Analysis of BCR-ABL neddylation levels in excised tumor xenografts from *Figure 1H*. (**N**) Verification of BCR-ABL neddylation sites in HEK293T cells transfected with indicated constructs.

The online version of this article includes the following source data and figure supplement(s) for figure 2:

**Source data 1.** Original file for the Western blot analysis in *Figure 2A*.

**Source data 2.** PDF containing *Figure 2A* and original scan of the relevant Western blot analysis with highlighted bands and sample labels.

**Source data 3.** Original file for the Western blot analysis in *Figure 2B*.

**Source data 4.** PDF containing *Figure 2B* and original scan of the relevant Western blot analysis with highlighted bands and sample labels.

**Source data 5.** Original file for the Western blot analysis in *Figure 2C*.

**Source data 6.** PDF containing *Figure 2C* and original scan of the relevant Western blot analysis with highlighted bands and sample labels.

**Source data 7.** Original file for the Western blot analysis in *Figure 2D*.

**Source data 8.** PDF containing *Figure 2D* and original scan of the relevant Western blot analysis with highlighted bands and sample labels.

**Source data 9.** Original file for the Western blot analysis in *Figure 2E*.

**Source data 10.** PDF containing *Figure 2E* and original scan of the relevant Western blot analysis with highlighted bands and sample labels.

**Source data 11.** Original file for the Western blot analysis in *Figure 2F*.

**Source data 12.** PDF containing *Figure 2F* and original scan of the relevant Western blot analysis with highlighted bands and sample labels.

**Source data 13.** Original file for the Western blot analysis in *Figure 2G*.

**Source data 14.** PDF containing *Figure 2G* and original scan of the relevant Western blot analysis with highlighted bands and sample labels.

**Source data 15.** Original file for the Western blot analysis in *Figure 2H*.

**Source data 16.** PDF containing *Figure 2H* and original scan of the relevant Western blot analysis with highlighted bands and sample labels.

**Source data 17.** Original file for the Western blot analysis in *Figure 2I*.

**Source data 18.** PDF containing *Figure 2I* and original scan of the relevant Western blot analysis with highlighted bands and sample labels.

**Source data 19.** Original file for the Western blot analysis in *Figure 2J*.

**Source data 20.** PDF containing *Figure 2J* and original scan of the relevant Western blot analysis with highlighted bands and sample labels.

**Source data 21.** Original file for the Western blot analysis in *Figure 2K*.

**Source data 22.** PDF containing *Figure 2K* and original scan of the relevant Western blot analysis with highlighted bands and sample labels.

**Source data 23.** Original file for the Western blot analysis in *Figure 2L*.

**Source data 24.** PDF containing *Figure 2L* and original scan of the relevant Western blot analysis with highlighted bands and sample labels.

**Source data 25.** Original file for the Western blot analysis in *Figure 2M*.

**Source data 26.** PDF containing *Figure 2M* and original scan of the relevant Western blot analysis with highlighted bands and sample labels.

**Source data 27.** Original file for the Western blot analysis in *Figure 2N*.

**Source data 28.** PDF containing *Figure 2N* and original scan of the relevant Western blot analysis with highlighted bands and sample labels.

**Figure supplement 1.** RAPSYN is an E3 ligase to neddylate BCR-ABL.

**Figure supplement 1—source data 1.** Original file for the Western blot analysis in *Figure 2—figure supplement 1A*.

**Figure supplement 1—source data 2.** PDF containing *Figure 2—figure supplement 1A* and original scan of the relevant Western blot analysis with highlighted bands and sample labels.

**Figure supplement 1—source data 3.** Original file for the Western blot analysis in *Figure 2—figure supplement 1B*.

**Figure supplement 1—source data 4.** PDF containing *Figure 2—figure supplement 1B* and original scan of the relevant Western blot analysis with highlighted bands and sample labels.

**Figure supplement 1—source data 5.** Original file for the Western blot analysis in *Figure 2—figure supplement 1D*.

**Figure supplement 1—source data 6.** PDF containing *Figure 2—figure supplement 1D* and original scan of the relevant Western blot analysis with highlighted bands and sample labels.

*Figure 2 continued*

**Figure supplement 1—source data 7.** Original file for the Western blot analysis in *Figure 2—figure supplement 1E*.

**Figure supplement 1—source data 8.** PDF containing *Figure 2—figure supplement 1E* and original scan of the relevant Western blot analysis with highlighted bands and sample labels.

**Figure supplement 1—source data 9.** Original file for the Western blot analysis in *Figure 2—figure supplement 1F*.

**Figure supplement 1—source data 10.** PDF containing *Figure 2—figure supplement 1F* and original scan of the relevant Western blot analysis with highlighted bands and sample labels.

**Figure supplement 1—source data 11.** Original file for the Western blot analysis in *Figure 2—figure supplement 1G*.

**Figure supplement 1—source data 12.** PDF containing *Figure 2—figure supplement 1G* and original scan of the relevant Western blot analysis with highlighted bands and sample labels.

**Figure supplement 1—source data 13.** Original file for the Western blot analysis in *Figure 2—figure supplement 1H*.

**Figure supplement 1—source data 14.** PDF containing *Figure 2—figure supplement 1H* and original scan of the relevant Western blot analysis with highlighted bands and sample labels.

**Figure supplement 1—source data 15.** Original file for the Western blot analysis in *Figure 2—figure supplement 1I*.

**Figure supplement 1—source data 16.** PDF containing *Figure 2—figure supplement 1I* and original scan of the relevant Western blot analysis with highlighted bands and sample labels.

**Figure supplement 2.** Liquid chromatography–mass spectrometry (LC–MS/MS) of trypsin-digested peptide fragments of neddylated BCR-ABL.

Next, we studied whether RAPSYN could directly mediate BCR-ABL neddylation. Strong BCR-ABL neddylation could be detected in all Ph$^+$ leukemia cell lines (*Figure 2E*, *Figure 2—figure supplement 1E*). More importantly, the neddylation of BCR-ABL was validated by immunoprecipitation using the PBMCs from two CML patients (*Figure 2F*). Treatment with the NAE1 inhibitor, MLN4924, significantly dampened the neddylation of BCR-ABL (*Figure 2G*, *Figure 2—figure supplement 1F*). In addition, the mutation of two glycine residues at the C-terminus of NEDD8 required for its covalent conjugating ability, or the co-expression of NEDP1 (NEDD8-specific protease 1) essentially diminished the neddylation of BCR-ABL (*Figure 2H, I*). As shown in *Figure 2J*, we co-expressed either WT-RAPSYN or its C366A mutant along with BCR-ABL and NEDD8, revealing that mutation of Cys to Ala at the catalytic residue C366 significantly decreased the neddylation level of BCR-ABL. Additionally, knockout of *RAPSN* abrogated BCR-ABL neddylation in the cells, and this effect was restored by transduction of *RAPSN* cDNA (*Figure 2K*). These results were further corroborated by in vitro experiments, which showed that BCR-ABL could hardly be neddylated in the absence of RAPSYN (*Figure 2L*). Consistently, the amount of neddylated BCR-ABL was markedly reduced in tumors generated by K562 cells transfected with sh*RAPSN*#3 (*Figure 2M*), indicating an essential role of the ligase activity of RAPSYN in BCR-ABL neddylation.

In addition, neither overall BCR-ABL expression nor its neddylation levels were affected after the knockdown of AChRs (*Figure 2—figure supplement 1*). Similarly, modulation of AChR activities and their downstream PKC–RAS–ERK and JAK2–AKT signaling pathways (*Kawamata et al., 2011*; *Aydın et al., 2013*) by either an agonist (carbachol) (*Jakubík et al., 2008*) or antagonists (benzethonium and homatropine) (*Durieux and Nietgen, 1997*) did not alter the expression or neddylation status of BCR-ABL (*Figure 2—figure supplement 1H, I*).

Subsequently, we tried to identified specific modification sites on BCR-ABL. The purified proteins were used for in vitro neddylation reactions, and the target bands were digested with trypsin for liquid chromatography–mass spectrometry (LC–MS/MS) analyses. Eight lysine residues were found to be potential NEDD8 accepting sites in BCR-ABL (*Figure 2—figure supplement 2*). To confirm these modification sites, a series of individual Lys-to-Arg mutants were generated. Except for K257, neddylation levels of BCR-ABL at other candidate sites were all significantly reduced, confirming the modification sites of these Lys residues (*Figure 2N*).

## RAPSYN attenuated c-CBL-mediated BCR-ABL ubiquitination and degradation

As decreased neddylation of BCR-ABL following either MLN4924 treatment or *RAPSN* KO was accompanied by a strong decline in its overall protein expression level (*Figures 2G and 3A, B*, *Figure 2—figure supplement 1F*, and *Figure 3—figure supplement 1*), we asked whether RAPSYN-mediated

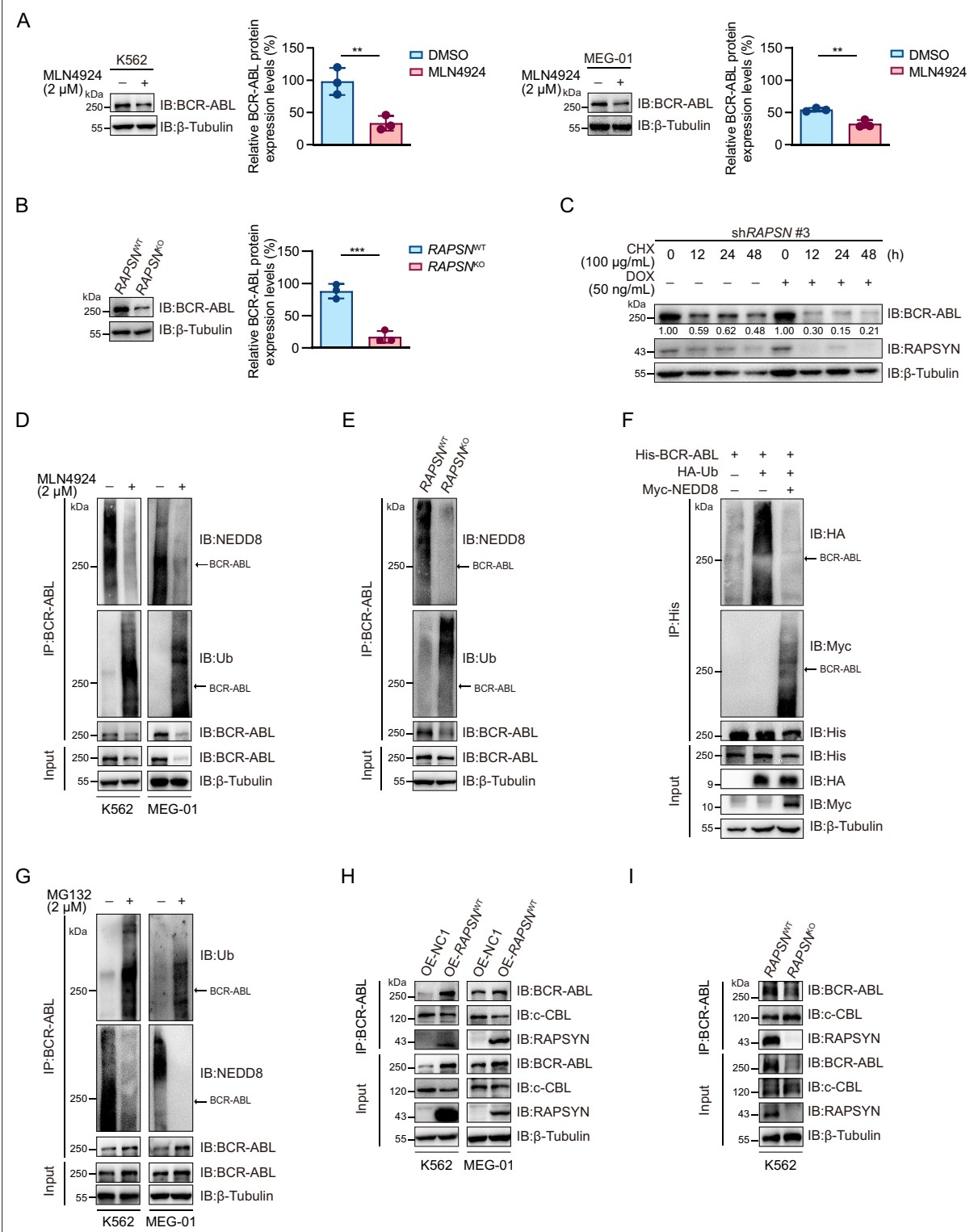

**Figure 3.** RAPSYN attenuates BCR-ABL ubiquitination and degradation. (**A**) Immunoblots of BCR-ABL in leukemic cells treated with MLN4924 or DMSO for 24 hr and corresponding quantification of three independent replicates. (**B**) Immunoblots of BCR-ABL in K562 WT and *RAPSN* KO cells and corresponding quantification of three independent replicates. (**C**) Assessment of BCR-ABL protein stability in K562 cells expressing DOX-inducible sh*RAPSN* #3 treated with CHX alone or in combination with DOX at indicated time points by immunoblotting. (**D**) Analysis of BCR-ABL neddylation and ubiquitination levels in leukemic cells treated with MLN4924 or DMSO for 24 hr. (**E**) Analysis of BCR-ABL neddylation and ubiquitination levels in K562 WT and *RAPSN* KO cells. (**F**) Immunoblots of HA and Myc after His-immunoprecipitation in HEK293T cells transfected with His-tagged BCR-ABL,

*Figure 3 continued on next page*

*Figure 3 continued*

HA-tagged Ub, or without Myc-tagged NEDD8. (**G**) Analysis of BCR-ABL ubiquitination and neddylation in leukemic cells treated with MG132 or DMSO for 12 hr. (**H**) Co-immunoprecipitation of BCR-ABL, c-CBL, and RAPSYN in leukemic cells expressing exogenous *RAPSN* cDNA or empty vector. (**I**) Co-immunoprecipitation of BCR-ABL, c-CBL, and RAPSYN in K562 WT and *RAPSN* KO cells. All data represent mean ± standard deviation (SD) of at least three independent experiments. p values were calculated using unpaired Student's *t*-test. **p < 0.01, ***p < 0.001.

The online version of this article includes the following source data and figure supplement(s) for figure 3:

**Source data 1.** Original file for the Western blot analysis in *Figure 3A*.

**Source data 2.** PDF containing *Figure 3A* and original scan of the relevant Western blot analysis with highlighted bands and sample labels.

**Source data 3.** Original file for the Western blot analysis in *Figure 3B*.

**Source data 4.** PDF containing *Figure 3B* and original scan of the relevant Western blot analysis with highlighted bands and sample labels.

**Source data 5.** Original file for the Western blot analysis in *Figure 3C*.

**Source data 6.** PDF containing *Figure 3C* and original scan of the relevant Western blot analysis with highlighted bands and sample labels.

**Source data 7.** Original file for the Western blot analysis in *Figure 3D*.

**Source data 8.** PDF containing *Figure 3D* and original scan of the relevant Western blot analysis with highlighted bands and sample labels.

**Source data 9.** Original file for the Western blot analysis in *Figure 3E*.

**Source data 10.** PDF containing *Figure 3E* and original scan of the relevant Western blot analysis with highlighted bands and sample labels.

**Source data 11.** Original file for the Western blot analysis in *Figure 3F*.

**Source data 12.** PDF containing *Figure 3F* and original scan of the relevant Western blot analysis with highlighted bands and sample labels.

**Source data 13.** Original file for the Western blot analysis in *Figure 3G*.

**Source data 14.** PDF containing *Figure 3G* and original scan of the relevant Western blot analysis with highlighted bands and sample labels.

**Source data 15.** Original file for the Western blot analysis in *Figure 3H*.

**Source data 16.** PDF containing *Figure 3H* and original scan of the relevant Western blot analysis with highlighted bands and sample labels.

**Source data 17.** Original file for the Western blot analysis in *Figure 3I*.

**Source data 18.** PDF containing *Figure 3I* and original scan of the relevant Western blot analysis with highlighted bands and sample labels.

**Figure supplement 1.** RAPSYN promotes BCR-ABL stabilization.

**Figure supplement 1—source data 1.** Original file for the Western blot analysis in *Figure 3—figure supplement 1*.

**Figure supplement 1—source data 2.** PDF containing *Figure 3—figure supplement 1* and original scan of the relevant Western blot analysis with highlighted bands and sample labels.

BCR-ABL neddylation affects protein stability. Subsequently, the protein synthesis inhibitor CHX was applied to K562 cells transduced with vectors encoding doxycycline-inducible *RAPSN* shRNA #3. Indeed, the expression levels of BCR-ABL declined much faster in cells with the induction of shRNA expression (*Figure 3C*). Meanwhile, we found that a clear inverse correlation between the neddylation and ubiquitination levels of BCR-ABL was observed (*Figure 3D, E*). BCR-ABL ubiquitination was remarkably reduced in the cells transfected with NEDD8 (*Figure 3F*). Consistent with these results, treatment of the cells with the proteasome inhibitor MG132 significantly increased the amount of ubiquitinated BCR-ABL accompanied by the decrease of BCR-ABL neddylation (*Figure 3G*).

To clarify the molecular basis of the battle between BCR-ABL neddylation and its ubiquitination, we detected that whether RAPSYN competes for binding to BCR-ABL with c-CBL, a reported E3 ligase mediating BCR-ABL ubiquitin–proteasome degradation (*Mao et al., 2010*). As a result, exogenous expression of *RAPSN* interfered with the interactions between BCR-ABL and c-CBL, whereas *RAPSN* ablation in K562 cells promoted c-CBL binding to BCR-ABL (*Figure 3H–I*). These data indicated that RAPSYN competes with c-CBL for binding to BCR-ABL, leading to subsequent BCR-ABL neddylation to enhance BCR-ABL stability by counteracting its proteasomal degradation.

## SRC-mediated phosphorylation stabilized RAPSYN by repressing its proteasomal degradation

SRC-family protein tyrosine kinases are capable of phosphorylating RAPSYN in neuronal system, among which SRC exerts the strongest function (*Mohamed and Swope, 1999*). In addition, SRC has been shown to be highly expressed in primary CML cells (*Yang et al., 2017*). We then studied whether SRC acts as an upstream regulator to mediate RAPSYN. SRC inhibition with saracatinib or shRNA not

only significantly downregulated phosphorylated (Tyr418) SRC, but also inhibited the phosphorylation of endogenous RAPSYN, resulting in a substantial decline in its protein level, whereas heterologous expression of *SRC* increased RAPSYN phosphorylation (*Figure 4A–C*, *Figure 4—figure supplement 1A*). Furthermore, in vitro incubation with recombinant RAPSYN, SRC, and ATP resulted in strong phosphorylation of RAPSYN, which could be fully abrogated by saracatinib treatment (*Figure 4D*). LC–MS/MS analyses indicated that Tyr residues at positions 59, 152, and 336 in RAPSYN are potential phosphorylation sites by SRC (*Figure 4—figure supplement 1B*). Then, after mutagenesis of these sites from Tyr to Phe, Y336, an evolutionarily conserved Tyr residue, was confirmed to be the primary site of RAPSYN phosphorylation (*Figure 4E*, *Figure 4—figure supplement 1C*). As SRC has no effect on *RAPSN* mRNA levels (*Figure 4—figure supplement 1D, E*), implying that SRC-mediated phosphorylation also affects RAPSYN stability. In fact, Ph$^+$ leukemia cells were treated with saracatinib, *SRC* silencing or mutation of the key phosphorylation site significantly accelerated the diminishment of RAPSYN expression following CHX treatment, conversely, expressing exogenous *SRC* cDNA prolonged the half-life of RAPSYN (*Figure 4F–I*).

To explore the molecular mechanisms responsible for the increased stability of phosphorylated RAPSYN, Ph$^+$ leukemia cells were treated with saracatinib or transduction of sh*SRC* followed by incubation with MG132. In all circumstances, MG132 could rescue the decrease of RAPSYN induced by saracatinib treatment or sh*SRC* knockdown (*Figure 4J, K*). Clearly, the specific phosphorylation of RAPSYN at Y336 by SRC led to its increased stability by preventing the proteasomal degradation, thereby maintaining the high levels of RAPSYN in Ph$^+$ leukemia.

## Phosphorylated RAPSYN potentiated its NEDD8 E3 ligase activity and promoted BCR-ABL stabilization

To dissect the role of SRC-mediated phosphorylation of RAPSYN, we tested whether phosphorylation of RAPSYN at Y336 affects its ligase activity. Immunoblotting revealed saracatinib treatment or *SRC* silencing reduced BCR-ABL neddylation and its protein expression, while exogenous expression of *SRC* cDNA strongly increased it (*Figure 5A–C*). Additionally, co-expression in HEK293T cells showed that Y336F mutation had no impact on BCR-ABL neddylation compared to co-transfection with SRC (*Figure 5D*). These results were supported by stronger neddylation of endogenous BCR-ABL in cells overexpressing WT *RAPSN*, but not in the Y336F mutant (*Figure 5E*). Furthermore, protein turnover rates of BCR-ABL were determined in Ph$^+$ leukemia cells and the cells with exogenous *RAPSN*$^{Y336F}$ expression displayed larger decrease in BCR-ABL level than those expressing *RAPSN*$^{WT}$ (*Figure 5F*). Therefore, RAPSYN phosphorylation at Y336 by SRC was a major contributing factor to its NEDD8 E3 ligase activity and BCR-ABL stability in Ph$^+$ leukemia cells.

## Phosphorylation of RAPSYN at Y336 promoted Ph$^+$ leukemia progression

To assess the extent to which SRC-mediated phosphorylation of RAPSYN at Y336 contributes to the enhanced viability of RAPSYN-dependent Ph$^+$ leukemia cells, we first identified a specific sh*SRC* by screening five candidates using toxicity tests and then performing rescue experiments with *SRC* cDNA. Toxicity tests revealed that, albeit to varying degrees, sh*SRC* #2-, #4-, and #5-induced cytotoxicity in all Ph$^+$ leukemia cell lines (*Figure 6A*, *Figure 6—figure supplement 1A*). However, exogenous *SRC* cDNA expression only restored the growth of Ph$^+$ leukemia cells transduced with sh*SRC* #2 (*Figure 6B*, *Figure 6—figure supplement 1B, C*).

Subsequently, we performed rescue experiments by introducing *RAPSN*$^{WT/Y336F}$ cDNA or an empty vector into sh*SRC* #2-transduced Ph$^+$ leukemia cells and found that virtually complete *RAPSN*$^{WT}$-induced rescue was detected in both cell lines, but *RAPSN*$^{Y336F}$ exhibited no restoring effect (*Figure 6C, D*). In addition, transduction of *RAPSN*$^{WT}$ cDNA conferred an increased resistance against saracatinib treatment, whereas the expression of *RAPSN*$^{Y336F}$ cDNA did not affect the drug sensitivity of the cells (*Figure 6E, F*). Furthermore, knockdown of *RAPSN* substantially sensitized both cell lines to saracatinib (*Figure 6G*).

In animal models, while the overall survival of mice intravenously injected with K562 cells expressing control empty vector was significantly improved by either saracatinib administration or shRNA-mediated SRC inhibition, overexpression of WT *RAPSN* fully counteracted these effects and shortened the lifespan of mice to the levels comparable to those of mice injected with K562 cells

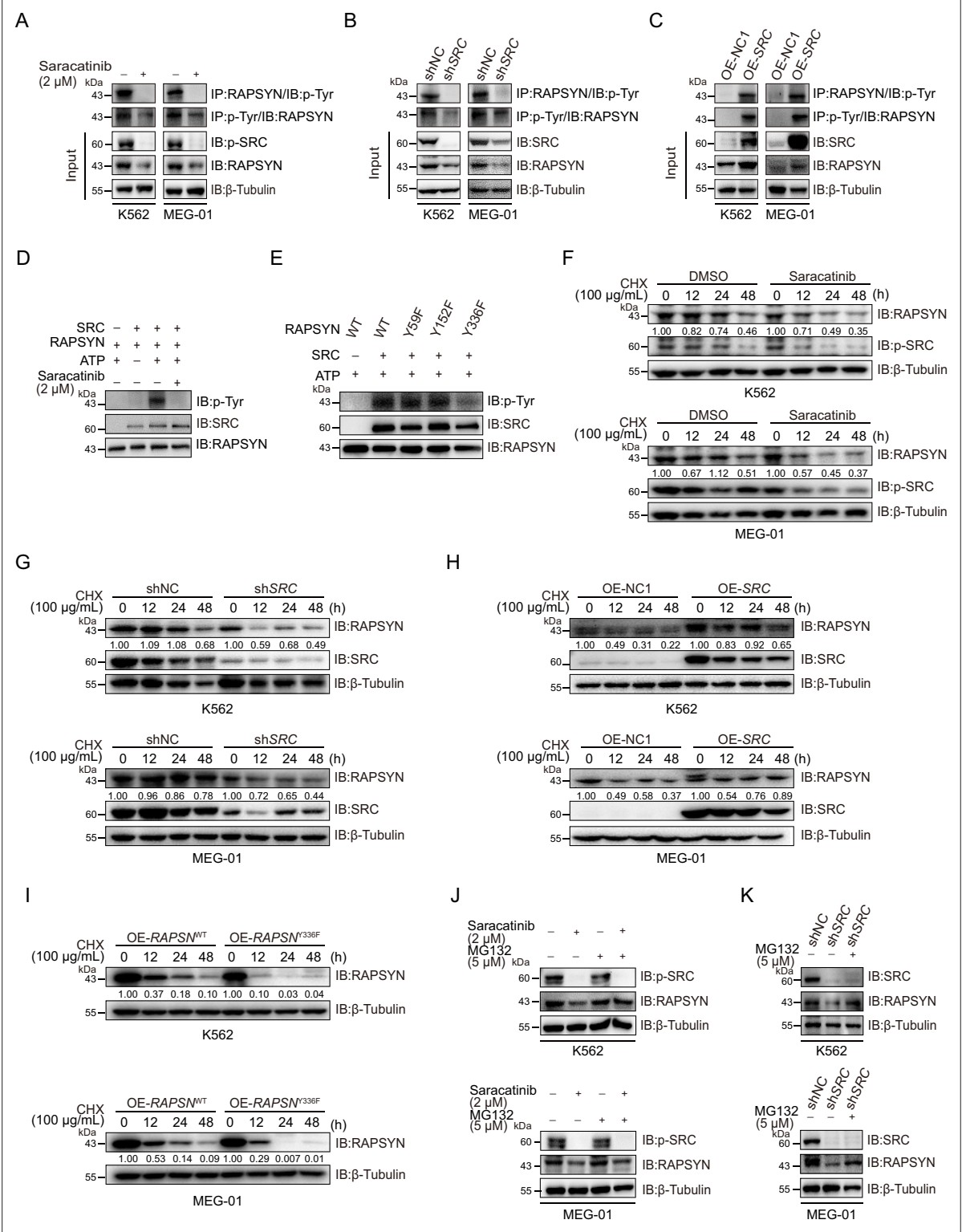

**Figure 4.** SRC-mediated phosphorylation at Y336 promotes RAPSYN stability by repressing its proteasomal degradation. (**A**) Assessment of RAPSYN phosphorylation levels in leukemic cells treated with saracatinib or DMSO for 24 hr. (**B**) Assessment of RAPSYN phosphorylation levels in leukemic cells transduced with sh*SRC* or shNC. (**C**) Assessment of RAPSYN phosphorylation levels in leukemic cells expressing exogenous *SRC* cDNA or empty vector. (**D**) Assessment of RAPSYN phosphorylation by SRC in vitro. Purified RAPSYN and SRC were incubated with ATP in the presence or absence of saracatinib for phosphorylation assay. (**E**) Verification of RAPSYN phosphorylation sites. Purified SRC and RAPSYN WT or indicated mutants were incubated with ATP for phosphorylation assay. (**F**) Assessment of RAPSYN protein stability in leukemic cells treated with CHX in combination with

*Figure 4 continued on next page*

*Figure 4 continued*

saracatinib or DMSO at indicated time points by immunoblotting. (**G**) Assessment of RAPSYN protein stability in leukemic cells transduced with sh*SRC* or shNC by immunoblotting. (**H**) Assessment of RAPSYN protein stability in leukemic cells transduced with exogenous *SRC* cDNA or empty vector by immunoblotting. (**I**) Assessment of RAPSYN protein stability in leukemic cells transduced with exogenous *RAPSN* WT or Y336F cDNA by immunoblotting. (**J**) Immunoblots of RAPSYN in leukemic cells treated with saracatinib or DMSO for 12 hr, and subsequently with MG132 or DMSO for another 12 hr. (**K**) Immunoblots of RAPSYN in leukemic cells transduced with shNC or sh*SRC* and treated with MG132 or DMSO for 12 hr.

The online version of this article includes the following source data and figure supplement(s) for figure 4:

**Source data 1.** Original file for the Western blot analysis in *Figure 4A*.

**Source data 2.** PDF containing *Figure 4A* and original scan of the relevant Western blot analysis with highlighted bands and sample labels.

**Source data 3.** Original file for the Western blot analysis in *Figure 4B*.

**Source data 4.** PDF containing *Figure 4B* and original scan of the relevant Western blot analysis with highlighted bands and sample labels.

**Source data 5.** Original file for the Western blot analysis in *Figure 4C*.

**Source data 6.** PDF containing *Figure 4C* and original scan of the relevant Western blot analysis with highlighted bands and sample labels.

**Source data 7.** Original file for the Western blot analysis in *Figure 4D*.

**Source data 8.** PDF containing *Figure 4D* and original scan of the relevant Western blot analysis with highlighted bands and sample labels.

**Source data 9.** Original file for the Western blot analysis in *Figure 4E*.

**Source data 10.** PDF containing *Figure 4E* and original scan of the relevant Western blot analysis with highlighted bands and sample labels.

**Source data 11.** Original file for the Western blot analysis in *Figure 4F*.

**Source data 12.** PDF containing *Figure 4F* and original scan of the relevant Western blot analysis with highlighted bands and sample labels.

**Source data 13.** Original file for the Western blot analysis in *Figure 4G*.

**Source data 14.** PDF containing *Figure 4G* and original scan of the relevant Western blot analysis with highlighted bands and sample labels.

**Source data 15.** Original file for the Western blot analysis in *Figure 4H*.

**Source data 16.** PDF containing *Figure 4H* and original scan of the relevant Western blot analysis with highlighted bands and sample labels.

**Source data 17.** Original file for the Western blot analysis in *Figure 4I*.

**Source data 18.** PDF containing *Figure 4I* and original scan of the relevant Western blot analysis with highlighted bands and sample labels.

**Source data 19.** Original file for the Western blot analysis in *Figure 4J*.

**Source data 20.** PDF containing *Figure 4J* and original scan of the relevant Western blot analysis with highlighted bands and sample labels.

**Source data 21.** Original file for the Western blot analysis in *Figure 4K*.

**Source data 22.** PDF containing *Figure 4K* and original scan of the relevant Western blot analysis with highlighted bands and sample labels.

**Figure supplement 1.** SRC-mediated phosphorylation at Y336 promotes RAPSYN stability.

**Figure supplement 1—source data 1.** Original file for the Western blot analysis in *Figure 4—figure supplement 1A*.

**Figure supplement 1—source data 2.** PDF containing *Figure 4—figure supplement 1A* and original scan of the relevant Western blot analysis with highlighted bands and sample labels.

without SRC inhibition. In contrast, expression of exogenous *RAPSN*^Y336F attenuated the protective effects of SRC inhibition to a much lesser extent (*Figure 6H–J*). Taken together, these results suggest that phosphorylation at Y336 by SRC is a major event in the pro-leukemogenic functions of RAPSYN in Ph[+] leukemia development.

## Discussion

In this study, we identified a novel role of RAPSYN in hematology. Indeed, RAPSYN inhibition significantly suppressed the survival of Ph[+] leukemia. This phenotype was found to be linked to the NEDD8 E3 ligase activity of RAPSYN, which mediated BCR-ABL neddylation to enhance its stability for promoting leukemogenesis.

Balanced protein synthesis and degradation are pivotal for maintaining protein homeostasis and normal cellular function. Neddylation is a type of important PTMs that mainly modulates protein stability. Accumulating evidence has shown that targeting the neddylation process could be an appealing strategy for anticancer therapy, with a particular efficacy shown in hematologic malignancies (*Bhalla and Fiskus, 2016*; *McGrail et al., 2020*; *Norton et al., 2021*; *Xie et al., 2021*). On the other hand, neddylation is a specific PTM that modifies multiple Lys residues in BCR-ABL, shielding

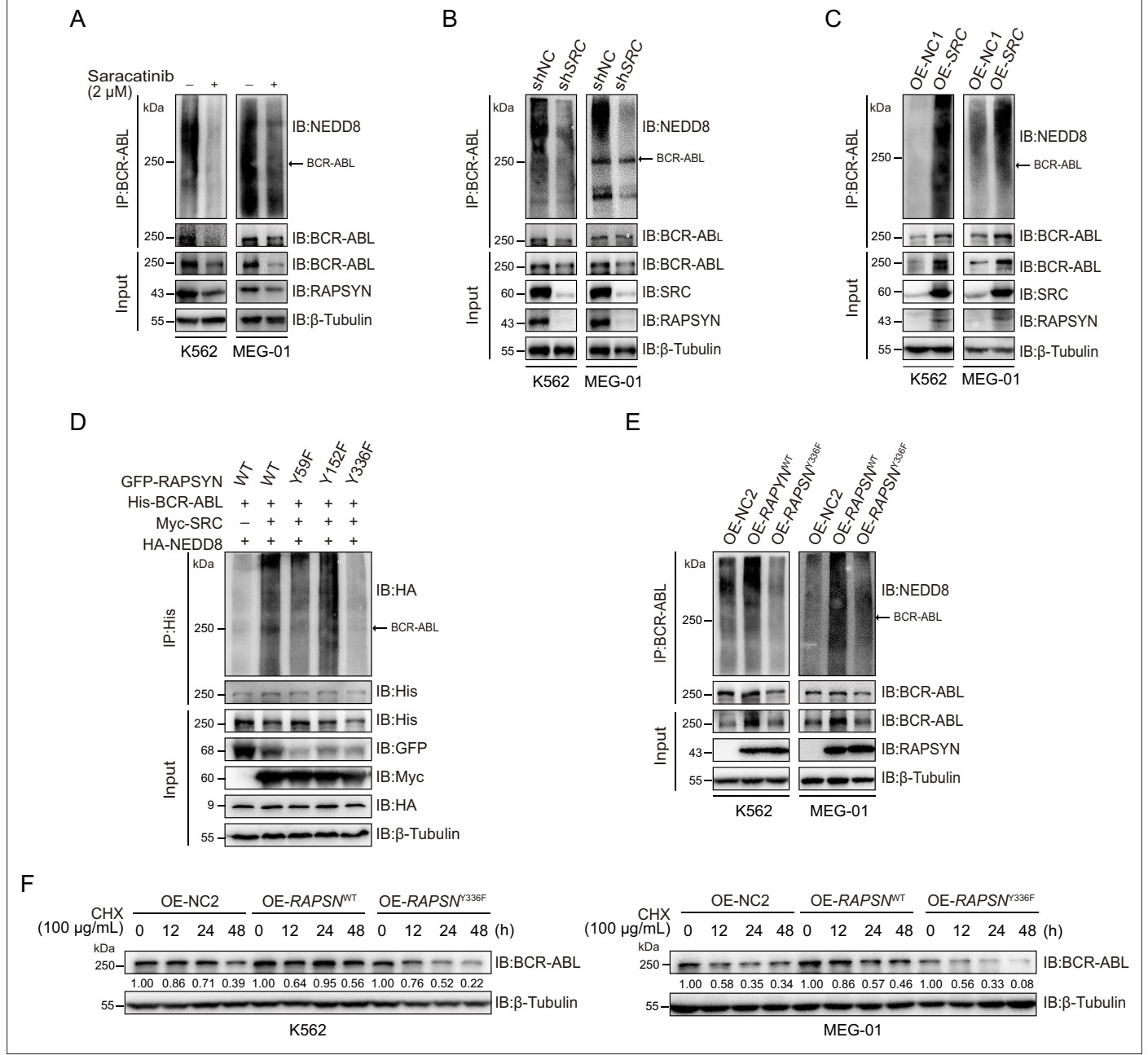

**Figure 5.** RAPSYN phosphorylation at Y336 potentiates its E3 ligase activity and promotes BCR-ABL stabilization. (**A**) Immunoblots of BCR-ABL neddylation levels in leukemic cells treated with saracatinib or DMSO for 24 hr. (**B**) Immunoblots of BCR-ABL neddylation levels in leukemic cells transduced with sh*SRC* or shNC. (**C**) Immunoblots of BCR-ABL neddylation levels in leukemic cells expressing exogenous *SRC* cDNA or empty vector. (**D**) Effects of RAPSYN phosphorylation on BCR-ABL neddylation levels in HEK293T cells transfected with indicated constructs. (**E**) Effects of RAPSYN phosphorylation at Y336 on BCR-ABL neddylation levels in leukemic cells expressing exogenous *RAPSN* WT, Y336F cDNA, or empty vector. (**F**) Assessment of BCR-ABL protein stability in leukemic cells transduced with exogenous cDNA for *RAPSN*-WT, Y336F mutant or empty vector by immunoblotting.

The online version of this article includes the following source data for figure 5:

**Source data 1.** Original file for the Western blot analysis in *Figure 5A*.

**Source data 2.** PDF containing *Figure 5A* and original scan of the relevant Western blot analysis with highlighted bands and sample labels.

**Source data 3.** Original file for the Western blot analysis in *Figure 5B*.

**Source data 4.** PDF containing *Figure 5B* and original scan of the relevant Western blot analysis with highlighted bands and sample labels.

**Source data 5.** Original file for the Western blot analysis in *Figure 5C*.

*Figure 5 continued on next page*

*Figure 5 continued*

**Source data 6.** PDF containing *Figure 5C* and original scan of the relevant Western blot analysis with highlighted bands and sample labels.

**Source data 7.** Original file for the Western blot analysis in *Figure 5D*.

**Source data 8.** PDF containing *Figure 5D* and original scan of the relevant Western blot analysis with highlighted bands and sample labels.

**Source data 9.** Original file for the Western blot analysis in *Figure 5E*.

**Source data 10.** PDF containing *Figure 5E* and original scan of the relevant Western blot analysis with highlighted bands and sample labels.

**Source data 11.** Original file for the Western blot analysis in *Figure 5F*.

**Source data 12.** PDF containing *Figure 5F* and original scan of the relevant Western blot analysis with highlighted bands and sample labels.

---

this oncoprotein to compete ubiquitination-mediated degradation, which provides a reasonable explanation on the poor in vivo efficacy of PROTAC-based degraders for BCR-ABL (*Li and Song, 2020*). MLN4924, a NEDD8-activating E1 enzyme inhibitor, has been shown to inhibit the survival of both wild-type (WT) and T315I-BCR-ABL leukemia cells as well as LICs (*Liu et al., 2018*; *Bahjat et al., 2019*; *Guo et al., 2019*). Moreover, clinical trials of MLN4924 in combination with anticancer agents in acute myeloid leukemia have progressed to phase II (NCT03745352 and PEVENAZA [NCT04266795]) and III (PANTHER [NCT03268954] and PEVOLAM [NCT04090736]). However, the neddylation system works in a substrate- and context-dependent manner, which essentially defines its role in tumorigenesis as *anti* or *pro*, particularly relying on the substrate specificity of the NEDD8 E3 ligase. Neddylation can either facilitate ubiquitination-dependent degradation of its substrates, such as EGFR (*Oved et al., 2006*) and c-SRC (*Lee et al., 2018*), or enhance protein stability in the cases of HuR (*Embade et al., 2012*) and TGF-β type II receptor (*Zuo et al., 2013*). Thus, the antitumor effects of MLN4924 are the integrative outcome of inhibiting more *pro-* than *anti*-tumorigenic neddylation activities in the reported tumor types. Recent studies uncovered that neddylation could also inhibit tumor progression and MLN4924 stimulates tumor sphere formation and wound healing as well as promotes glycolysis (*Zhou et al., 2016*; *Zhou et al., 2019*; *Zhou and Sun, 2019*). Therefore, rather than suppressing the entire neddylation system to affect a wide range of proteins, targeted inhibition of substrate-specific NEDD8 E3 ligase, such as RAPSYN, might offer a potential therapeutic opportunity for more elegant anticancer intervention with fewer side effects.

Functional studies on RAPSYN have focused on its contribution to neuromuscular transmission (*Xing et al., 2019*; *Xing et al., 2020*). In this study, we found that RAPSYN promotes disease progression by neddylating BCR-ABL for its resistance to c-CBL-mediated proteasomal degradation. Additionally, its NEDD8 E3 ligase activity was increased by SRC-mediated phosphorylation. Previously, the residue Y86 in RAPSYN was identified as a phosphorylation site by muscle-associated receptor tyrosine kinase (MuSK) at the neuromuscular junction endplate (*Xing et al., 2019*), which could enhance the ligase activity of RAPSYN by mediating its self-association (*Xing et al., 2020*). Differently, we found that SRC phosphorylates RAPSYN at Y336 residue located between its CC and RING domains in Ph[+] leukemia cells, suggesting that the phosphorylation of RAPSYN might be kinase or tissue specific. So far, multiple RAPSYN mutations have been reported to be causative (*Cossins et al., 2006*; *Finsterer, 2019*). In particular, N88K mutation could significantly reduce MuSK-mediated Y86 phosphorylation of RAPSYN to affect its E3 NEDD8 ligase activity (*Xing et al., 2019*). However, the precise process by which N88K mutation is involved in modulating RAPSYN phosphorylation is unclear. On the other hand, N88 was predicted to be a *N*-glycosylation site with the highest score among all putative ones in RAPSYN (*Lam et al., 2017*), implicating that *N*-glycosylation of RAPSYN could be a prerequisite for normal RAPSYN phosphorylation and activation. In the present study, we unveiled that SRC-mediated RAPSYN phosphorylation could substantially potentiate its NEDD8 E3 ligase activity and enhance its protein stability, once again suggesting the hematological specificity. In addition, given the fact that neddylation and sumoylation have both shown, as also in the present study, to be capable of antagonizing the ubiquitination of their substrates (*Enchev et al., 2015*; *Yang et al., 2017*), the potential self-modification of RAPSYN is likely to promote its own stabilization. Collectively, it is of great importance to further dissect different PTMs of RAPSYN and their interactions in order to better understand RAPSYN's biological functions.

It is well known that the generation of BCR-ABL fusion protein is a decisive characteristic of Ph[+] leukemia (*de Klein et al., 1982*). Aside from the occurrence of RAPSYN in patients with Ph[+] leukemia, the present study revealed a fascinating finding that BCR-ABL expression levels were correlated with

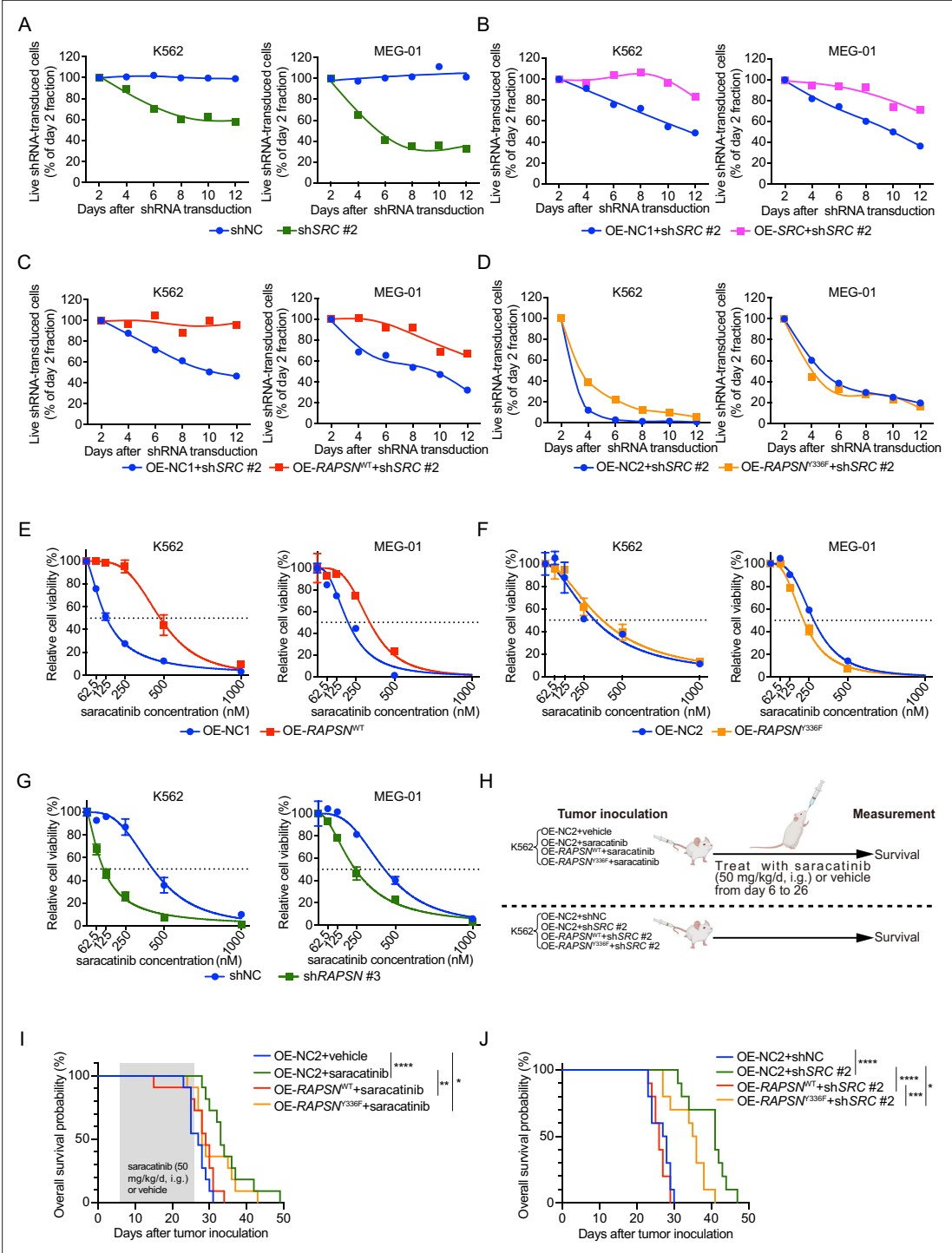

**Figure 6.** SRC-mediated phosphorylation of RAPSYN at Y336 promotes Ph⁺ leukemia progression. (**A**) Cytotoxicity induced by sh*SRC* #2-mediated *SRC* knockdown in leukemic cells. (**B**) Rescue of leukemic cells from sh*SRC* #2-induced toxicity by exogenous expression of *SRC* cDNA. (**C**) Rescue of leukemic cells from sh*SRC* #2-induced toxicity by exogenous expression of *RAPSN*^WT cDNA. (**D**) Failed rescue of leukemic cells from sh*SRC* #2-induced toxicity by exogenous expression of *RAPSN*^Y336F cDNA. (**E**) Viability of leukemic cells transduced with either *RAPSN*^WT cDNA or corresponding empty vector after 72 hr of incubation with indicated concentrations of saracatinib. (**F**) Viability of leukemic cells transduced with either *RAPSN*^Y336F cDNA or corresponding empty vector after 72 hr of incubation with indicated concentrations of saracatinib. (**G**) Viability of leukemic cells transduced with either shNC or sh*RAPSN* #3 after 72 hr of incubation with indicated concentrations of saracatinib. (**H**) Experimental design used to test in vivo effects of RAPSYN phosphorylation at Y336 on Ph⁺ leukemia progression and survival time. (**I**) Kaplan–Meier survival curve of NCG mice following intravenous injection of K562-*RAPSN*^WT or K562-*RAPSN*^Y336F cells and intragastric administration of saracatinib or corresponding vehicle from days 6 to 26 as

*Figure 6 continued on next page*

indicated (ten mice in each group). (**J**) Kaplan–Meier survival curve of NCG mice following intravenous injection of double-transfected K562 cells (ten mice in each group). Representative results from at least three independent experiments are shown (**A–G**); error bars, mean ± standard deviation (SD); *p < 0.05, **p < 0.01, ***p < 0.001, ****p < 0.0001; log-rank test (**I–J**).

The online version of this article includes the following figure supplement(s) for figure 6:

**Figure supplement 1.** *shSRC* #2 is a specific shRNA targeting the 3'UTR of SRC.

those of RAPSYN, demonstrating the specificity of RAPSYN-mediated neddylation of BCR-ABL. Moreover, as a new type of PTM for BCR-ABL, RAPSYN-mediated neddylation was found to compete c-CBL-mediated ubiquitination, causing a reduction in BCR-ABL degradation. Given the fact that the increase of BCR-ABL expression can affect the sensitivity to TKIs and eventually determine the rate of TKI resistance and LIC population in patients with Ph+ leukemia (*Issaad et al., 2000*; *Barnes et al., 2005*), effective degradation of BCR-ABL is an alternative opportunity for the treatment. Although recent studies have greatly advanced our understanding of the regulation of BCR-ABL degradation (*Burslem et al., 2019*; *Shibata et al., 2020*; *Jiang et al., 2021a*), most reported modulatory proteins are not ideal for translation to clinical settings because of their pivotal roles in sustaining normal hematological functions. In contrast, RAPSYN was nearly unexpressed in the blood of healthy donors. Thus, it is reasonable to expect that the inhibition of RAPSYN expression could lead to cytotoxicity in Ph+ leukemia with high specificity and marginal side effects. Furthermore, our present results showed that knockdown of RAPSYN significantly increased the sensitivity of leukemia cells to saracatinib, implying that a combination of RAPSYN inhibition and TKI treatment can effectively control mutations and LIC-derived TKI resistance in Ph+ leukemia.

In summary, our work has uncovered the pivotal role that RAPSYN exerts its NEDD8 E3 ligase activity in neddylating and stabilizing BCR-ABL in the pathogenesis of Ph+ leukemia and thus delineate it as a potential novel therapeutic target for the treatment of Ph+ leukemia. More importantly, our results shed a light on future investigations that may help to extend to other cancer types for broadening our understanding of RAPSYN's involvement in hematology and oncology.

# Materials and methods

## Key resources table

| Reagent type (species) or resource | Designation | Source or reference | Identifiers | Additional information |
|---|---|---|---|---|
| Strain, strain background (*M. musculus*) | Mouse: NOD/ShiLtJGpt-Prkdc<sup>em26Cd52</sup> Il2rg<sup>em26Cd22</sup>/Gpt | GemPharmatech | Cat# CB101 | |
| Cell line (*Homo sapiens*) | Human bone marrow stromal cell HS-5 (male) | ATCC | Cat# CRL11882; RRID:CVCL_3720 | |
| Cell line (*Homo sapiens*) | Human chronic myelogenous leukemia K562 (female) | COBIOER | Cat# CBP60529 | |
| Cell line (*Homo sapiens*) | Human chronic myelogenous leukemia MEG-01 (male) | COBIOER | Cat# CBP61104 | |
| Cell line (*Homo sapiens*) | Human chronic myelogenous leukemia KU812 (male) | COBIOER | Cat# CBP60732 | |
| Cell line (*Homo sapiens*) | HEK-293T | KeyGEN BioTECH | Cat# KG405 | |
| Antibody | Mouse monoclonal anti-RAPSYN (clone 1234) | Abcam | Cat# ab11423; RRID:AB_298028 | 1:1000 |
| Antibody | Rabbit polyclonal anti-RAPSYN (clone 118491) | Abcam | Cat# ab118491; RRID:AB_10899872 | 1:1000 |
| Antibody | Mouse monoclonal anti-6X His tag (clone HIS.H8) | Abcam | Cat# ab18184; RRID:AB_444306 | 1:1000 |

*Continued on next page*

*Continued*

| Reagent type (species) or resource | Designation | Source or reference | Identifiers | Additional information |
|---|---|---|---|---|
| Antibody | Mouse monoclonal anti-BCR-ABL (clone 7C6) | Abcam | Cat# ab187831 | 1:1000 |
| Antibody | Rabbit monoclonal anti-SRC Family (phosphoY418) | Abcam | Cat# ab40660 | 1:1000 |
| Antibody | Rabbit monoclonal anti-NEDD8 (clone 19E3) | Cell Signaling Technology | Cat# 2754; RRID:AB_659972 | 1:1000 |
| Antibody | Rabbit monoclonal anti-HA-Tag (clone C29F4) | Cell Signaling Technology | Cat# 3724; RRID:AB_1549585 | 1:1000 |
| Antibody | Rabbit monoclonal anti-GFP (clone D5.1) | Cell Signaling Technology | Cat# 2956; RRID:AB_1196615 | 1:1000 |
| Antibody | Rabbit monoclonal anti-GST (clone 91G1) | Cell Signaling Technology | Cat#2625 | 1:1000 |
| Antibody | Mouse monoclonal anti-Myc-Tag (clone 9B11) | Cell Signaling Technology | Cat#2276; RRID:AB_331783 | 1:1000 |
| Antibody | Rabbit monoclonal anti-SRC (clone 36D10) | Cell Signaling Technology | Cat# 2109; RRID:AB_2106059 | 1:1000 |
| Antibody | Rabbit monoclonal anti-Flag (DYKDDDDK) Tag (clone D6W5B) | Cell Signaling Technology | Cat# 14793; RRID:AB_2572291 | 1:1000 |
| Antibody | Rabbit monoclonal anti-GAPDH (clone 14C10) | Cell Signaling Technology | Cat# 2128; RRID:AB_823664 | 1:2000 |
| Antibody | Anti-mouse IgG, HRP-linked antibody | Cell Signaling Technology | Cat# 7076 S; RRID:AB_330924 | 1:5000 |
| Antibody | Anti-rabbit IgG, HRP-linked antibody | Cell Signaling Technology | Cat# 7074 S; RRID:AB_2099233 | 1:5000 |
| Antibody | Normal Mouse IgG | Santa Cruz Biotechnology | Cat# sc-2025; RRID:AB_737182 | 1:100 |
| Antibody | Mouse monoclonal anti-c-CBL (clone A-9) | Santa Cruz Biotechnology | Cat# SC-1651; RRID:AB_2244054 | 1:1000 |
| Antibody | Mouse monoclonal anti-Phosphotyrosine Antibody (clone 4G10) | Sigma-Aldrich | Cat# 05-321 X; RRID:AB_568858 | 1:1000 |
| Antibody | Rabbit monoclonal anti-AChRα7 | Santa Cruz Biotechnology | Cat# SC-58607; RRID:AB_784835 | 1:1000 |
| Antibody | Rabbit monoclonal anti-mAChR $M_2$ | Santa Cruz Biotechnology | Cat# SC-33712; RRID:AB_673789 | 1:1000 |
| Antibody | Mouse monoclonal anti-mAChR $M_3$ | Santa Cruz Biotechnology | Cat# SC-518107 | 1:1000 |
| Antibody | Rbbit polyclonal anti-mAChR M4 | HUABIO | Cat# ER1906-24; | 1:1000 |
| Strain, strain background (*Escherichia coli*) | DH5-alpha | TIANGEN | Cat# CB101 | |
| Strain, strain background (*Escherichia coli*) | ArcticExpress (DE3) pRARE2 | ANGYUBIO | Cat# AYBIO-G6023 | |
| Transfected construct (human) | Plasmid: pcDNA 3.1(+) mammalian expression vector | Invitrogen | Cat# V79020 | |
| Strain, strain background (human) | Plasmid: pd1-EGFP-N1 mammalian expression vector | Clontech | Cat# 6073-1 | |
| Transfected construct (*Escherichia coli*) | pGEX-4T-1 bacterial expression vector | Addgene | 27-4580-01 | |

## Human clinical samples

This study was approved by the ethics committee of the First Affiliated Hospital of Nanjing Medical University (2019-SR-485.A1). Human peripheral blood samples were obtained from the remaining material utilized for routine laboratory tests at the First Affiliated Hospital of Nanjing Medical University (Nanjing, China) and derived from 21 patients with Ph+ CML and six healthy volunteers. And one human bone marrow sample of Ph+ acute lymphoblastic leukemia patient was obtained from the remaining material utilized for routine laboratory tests at the First Affiliated Hospital of Nanjing Medical University (Nanjing, China). Peripheral blood and bone marrow mononuclear cells were isolated by density gradient centrifugation using the Ficoll Paque Plus solution (17-1440-02, GE Healthcare).

## Cell cultures

K562 (female; CBP60529), MEG-01 (male; BP61104), KU812 (male; BP60732), and Jurkat cells were purchased from COBIOER and cultured in Roswell Park Memorial Institute 1640 medium (RPMI 1640; KGM31800, KeyGEN BioTECH) containing 10–20% fetal bovine serum (FBS; FS301-02, TransGen Biotech) and 100 mg/ml streptomycin/penicillin (FG101-01, TransGen Biotech). HS-5 (male; CRL11882, ATCC), purchased from the American Type Culture Collection (ATCC) and HEK293T (KG405) from KeyGEN BioTECH were cultured in Dulbecco's modified Eagle's medium (KGM12800, KeyGEN BioTECH) containing 10% FBS and 100 mg/ml streptomycin/penicillin. All cells were cultured in a humidified incubator with 5% $CO_2$ at 37°C. All the cell lines were authenticated using short tandem repeat matching analysis and tested negative for mycoplasma contamination.

## Animal studies

Female NOD/ShiLtJGpt-Prkdcem26Cd52Il2rgem26Cd22/Gpt (NCG) mice (6–8 weeks), purchased from GemPharmatech Co, Ltd, were used for all in vivo studies. Mice were housed under specific pathogen-free conditions at 24 ± 1°C and 55 ± 5% humidity in a barrier facility with 12 hr light–dark cycles. All animal experiments were performed in accordance with the National Institutes of Health Guide for the Care and Use of Laboratory Animals with the approval of the Center for New Drug Evaluation and Research, China Pharmaceutical University (approval number: B20190925-1; Nanjing, China).

## Apoptosis assay

A total of 1–5 × 10^5 cells were washed with phosphate-buffered saline (PBS; 02-024-1ACS, Biological Industries) and resuspended in 100 µl of Annexin V binding buffer (E-CK-A211, Elabscience). The cell suspension was incubated with 2.5 µl of Annexin V-AF647 (E-CK-A211, Elabscience) and 2.5 µl of propidium iodide (PI; E-CK-A211, Elabscience) for 20 min in the dark, followed by the addition of 400 µl of Annexin V binding buffer and detection by flow cytometry (Thermo Attune NxT, MA, USA).

## Cell viability assay

Cells were seeded at a density of 5000 (K562) or 20,000 cells (MEG-01) per well in round-bottom 96-well plates and incubated with different concentrations of saracatinib (AZD0530, Selleck) or the corresponding amount of solvent for 72 hr. Cells were then transferred to flat-bottom 96-well plates for the determination of cell viability using the CCK-8 Cell Counting Kit (A311-01, Vazyme) following the manufacturer's instructions. Each experiment was performed at least three times for individual cell line.

## Cell proliferation assay

Cells were washed twice with PBS, resuspended with a cell number of 2 × 10^6 in 1 ml PBS, and incubated with 2.5 µM carboxyfluorescein succinimidyl ester (CFSE) solution (1948076, Thermo Scientific) or 5 µM carboxylic acid, acetate, and succinimidyl ester (SNARF-1) solution (S-22801, Invitrogen) for 20 min at 37°C in the dark, respectively. Subsequently, 1 ml of FBS was added to stop the staining, and the cells were washed twice with complete media. Cell division was monitored by measuring CFSE or SNARF-1 dilution using flow cytometry via channels BL1 and YL1.

## Cell cycle analysis

A total of 1 × 10^6 cells were washed once with pre-chilled PBS, fixed in ice-cold 70% ethanol, vortexing and kept for at least 20 min at −20°C. The fixed cells were washed twice with PBS and stained with

20 μg/ml PI containing 100 mg/ml RNAse A (740505, MACHEREY-NAGEL) for 15 min at room temperature. Stained nuclei were analyzed by flow cytometry and quantified using FlowJo software (BD Biosciences, NJ, USA).

### Cell transfection and viral transduction

Transfection of the indicated DNA plasmids into HEK293T cells was performed using Lipofectamine 2000 (11668500, Thermo Fisher Scientific), according to the manufacturer's instructions. Briefly, transfection of HEK293T was performed when cell confluency reached 60–70%. Plasmids and Lipofectamine 2000 reagent were diluted in Opti-MEM medium (31985-070, Thermo Fisher Scientific) and incubated for 5 min at room temperature. They were then mixed together, incubated for another 20 min at room temperature, and added to the target cells. The transfected cells were collected after 48 hr for further analysis.

### GST pull-down assay

Recombinantly expressed and purified GST or GST-RAPSYN (GenBank: NM_005055.5; 1236 bp open reading frame (ORF) sequence) proteins were incubated with glutathione beads 4FF (SA010010, Smart-Lifescience) overnight at 4°C in binding buffer (0.14 M NaCl, 2.68 mM KCl, 2 mM $KH_2PO_4$, 0.01 M $Na_2HPO_4$, 10 mM 1,4-dithiothreitol (DTT), pH 7.4), respectively, and incubated with purified His-BCR-ABL (p210 BCR-ABL (b3a2); 6126 bp ORF sequence) protein for another 4 hr at 4°C. Beads were washed three times with washing buffer (0.14 M NaCl, 2.68 mM KCl, 2 mM $KH_2PO_4$, 0.01 M $Na_2HPO_4$, 0.5 mM reduced glutathione (GSH), 10 mM DTT, pH 7.4), eluted with elution buffer (0.14 M NaCl, 2.68 mM KCl, 2 mM $KH_2PO_4$, 0.01 M $Na_2HPO_4$, 10 mM reduced GSH, 10 mM DTT, pH 7.4), and subjected to immunoblotting detection.

### Immunoblotting

The cells were lysed on ice with Nonidet P-40 (NP-40) lysis buffer (150 mM NaCl, 100 mM NaF, 50 mM Tris–HCl (pH 7.6), and 0.5% NP-40) supplemented with a protease inhibitor cocktail (78446, Thermo Fisher Scientific). Lysates were centrifuged, quantified, subjected to sodium dodecyl sulfate–polyacrylamide gel electrophoresis (SDS–PAGE), and transferred to polyvinylidene difluoride (PVDF) membranes using a Bio-Rad transfer apparatus. Membranes were blocked with 5% non-fat milk in Tris-buffered saline buffer containing 0.1% Tween-20 (TBST) at 20–25°C for 2 hr, followed by incubation with primary antibody overnight at 4°C. The membranes were then washed three times in TBST buffer and incubated with species-specific horseradish peroxidase (HRP)-conjugated secondary antibodies for 2 hr at room temperature. Then, the membranes were washed three times in TBST buffer, developed using the enhanced chemiluminescence (ECL) reagent, and exposed to the ChemiDoc Imaging System (Tanon, Shanghai, China). The antibodies for immunoblotting or immunoprecipitation are list in Key Resources Table.

### Immunoprecipitation

Immunoprecipitation assays were performed in accordance with the manufacturer's instructions. Briefly, the cells were lysed on ice with NP-40 lysis buffer supplemented with a protease inhibitor cocktail. Cell lysates were centrifuged, quantified, and incubated with the appropriate primary antibody overnight at 4°C, and subsequently with protein A agarose beads (16–125, Millipore) for another 4 hr at 4°C. Agarose was washed three times with lysis buffer and eluted with SDS–PAGE loading buffer. The eluted immunocomplexes were separated by SDS–PAGE and transferred to PVDF membranes. The membranes were then probed with primary and corresponding secondary antibodies, washed three times in TBST buffer, developed using ECL reagent and exposed by ChemiDoc Imaging System.

### In vitro neddylation assay

A 30-μl reaction mixture containing 2 mM ATP-$Mg^{2+}$ (B-20, R&D), 50 ng E1 (APPBP1/UBA3; E-313-25, R&D), 400 ng E2 (UBE2M; E2-656-100, R&D), 0.25 μg NEDD8 (UL-812-500, R&D), 0.35 μg His-BCR-ABL, with or without 4.77 μg recombinant RAPSYN was incubated at 37°C for 4 hr. Reaction was terminated with SDS–PAGE loading buffer and assayed using immunoblotting.

## In vitro phosphorylation assay

A 40-µl reaction mixture containing 2 mM ATP-Mg$^{2+}$, 3 µg recombinant RAPSYN, and 1 µg recombinant SRC (GenBank: NM_005417.5; 1608 bp ORF sequence) protein, with or without 2 µM saracatinib (AZD0530, Selleck), was incubated at 30°C for 30 min. Reaction was terminated with SDS–PAGE loading buffer and assayed by immunoblotting.

## Animal experiments with mouse models

Female NCG mice aged 6–8 weeks were used in all the animal experiments. In the subcutaneous tumor experiment shown in *Figure 1F*, 20 mice were randomly divided into two groups, followed by subcutaneous injection of $1 \times 10^6$ K562-shNC or K562-sh*RAPSN* #3 cells in 60 µl Matrigel (354234, Corning) into the right foreleg. Tumor size was measured every 2 days using a digital caliper. The tumor volume was quantified using the following equation: tumor volume = 0.5 × (long diameter) × (short diameter)$^2$. When the average volume of the control group exceeded 2000 mm$^3$, the mice were sacrificed. The tumors were separated, and their weights were measured. As shown in the survival experiment in *Figure 1K*, NCG mice were inoculated with K562-*RAPSN*$^{WT}$ or K562-*RAPSN*$^{KO}$ ($1 \times 10^7$ cells/mouse) via the tail vein. The survival time was recorded until the mice died. As shown in the survival experiment of *Figure 6I*, 40 mice were randomly divided into 4 groups and intravenously inoculated with K562-OE-NC2 (20 mice), K562-OE-*RAPSN*$^{WT}$ (10 mice), or K562-OE-*RAPSN*$^{Y336F}$ (10 mice). From days 6 to 26 after tumor cell inoculation, 10 mice inoculated with K562-OE-NC2 were treated with vehicle orally, while the other 30 mice inoculated with K562-OE-NC2, K562-OE-*RAPSN*$^{WT}$, or K562-OE-*RAPSN*$^{WT}$, 10 mice in each group, were administered saracatinib orally (50 mg/kg/day). The survival time was recorded until the mice died. In the survival experiment shown in *Figure 6J*, 40 mice were randomly divided into four groups and intravenously inoculated with double-transfected K562 cells, as indicated. The survival time was recorded until the mice died.

## Identification of modification sites

To determine which lysine residues in BCR-ABL were neddylated by NEDD8, an in vitro neddylation reaction (50 µl) was performed. After incubation at 37°C for 4 hr, the reaction mixture was separated by SDS–PAGE, and silver-stained bands were excised and sent to BiotechPack Scientific Co, Ltd (Beijing, China) for LC–MS/MS analysis. To determine which tyrosines in RAPSYN were phosphorylated by SRC, a phosphorylation reaction (50 µl) was performed. After a 30-min reaction, the reaction mixture was separated by SDS–PAGE, and silver-stained bands were excised and sent to Applied Protein Technology Co, Ltd (Shanghai, China) for LC–MS/MS analysis.

## Plasmid construction

Eukaryotic expression vectors encoding His-, GST-, HA-, Myc-, or Flag-tagged proteins were generated by inserting PCR-amplified fragments into the pcDNA3.1(+) mammalian expression vector (V79020, Invitrogen). Eukaryotic expression vectors encoding green fluorescent protein (GFP)-tagged proteins were generated by inserting PCR-amplified fragments into pd1-EGFP-N1 vector (6073-1, Clontech). Prokaryotic plasmids encoding GST-fusion proteins were constructed using pGEX-4T-1 bacterial expression vector (27-4580-01, Addgene). Mutants of His-, HA-, GST-, or GFP-tagged proteins were generated using QuickMutation Site-Directed Mutagenesis Kit (D0206, Beyotime) according to the manufacturer's instructions. Briefly, whole plasmid DNA was amplified by PCR for 20 cycles with specific mutant primers (*Supplementary file 1*) using QuickMutation site-directed mutagenesis kit. Next, 1 µl DpnI was directly added to the PCR reaction mixture, followed by incubation at 37°C for 30 min and transformation to *E. coli* cells. To verify the mutation sites, single colonies were selected for DNA sequencing and subsequent protein expression.

## Preparation of stable *RAPSYN*-KO K562 cell line

*RAPSN* KO K562 cells were generated using CRISPR/Cas9 system (Genloci Biotechnologies Inc). Single-guide RNAs for *RAPSN* (sgRNAs) were designed using online CRISPR design tool (http://crispr.mit.edu/). The sgRNA sequences were ATGGGGCGCTTCCGCGTGCTGGG, GTAGCGGCCCATCTCCGAGTGGG, and TCTGGTTGGACTGGTACAGCTGG, which were cloned into pGK1.1/CRISPR/Cas9 vector (Genloci Biotechnologies Inc). To obtain single clones of *RAPSN* KO cells, K562 cells were transfected with pGK1.1/CRISPR/Cas9 plasmid containing the aforementioned sgRNA sequence,

expanded, selected with puromycin (0120A21, LEAGENE), and isolated by single-cell culturing. Single clones obtained from *RAPSN* KO cells were validated by DNA sequencing and immunoblotting.

## Preparation of stable *RAPSN*-KD and *SRC*-KD cell lines

Lentivirus-producing shRNA targeting either human *RAPSN* or *SRC* mRNA was used to inhibit endogenous RAPSYN or SRC expression, respectively. All shRNAs (*Supplementary file 1*) were designed using online shRNA design tools (https://rnaidesigner.thermofisher.com and https://portals.broadinstitute.org). The shRNA primers were ordered from GenScript (Nanjing, China) and annealed in a thermal cycler according to following procedure (95°C, 2 min; 85°C, 9 min; 75°C, 9 min; 65°C, 9 min; 55°C, 9 min; 45°C, 9 min; 35°C, 9 min; 25°C, 10 min; 4°C, hold) in the presence of NE Buffer 2.1 (B7202S, New England BioLabs) to form a double strand with EcoRI and AgeI sticky ends. Using T4 DNA ligase (M0202L, NEW ENGLAND BioLabs), the double-stranded shRNAs were ligated with either lentiviral backbone plasmid vector pLKO-EGFP-puro or a tet pLKO-EGFP-puro, which was digested with the restriction enzymes EcoRI-HF (R3101S, NEW ENGLAND BioLabs) and AgeI-HF (R3552S, NEW ENGLAND BioLabs). Plasmids containing shRNA or corresponding empty vector were co-transfected with lentivirus packaging plasmids (pLP1, pLP2, and pLP/VSVG) into HEK293T cells using linear polyethylenimine (23966, Polyscience) transfection method. After transfection for 6–8 hr, the transfection reagent was replaced with a fresh medium. After incubation at 37°C, 5% $CO_2$ for 48 and 72 hr, the resulting lentivirus supernatant was collected, respectively, and filtrated through a 0.22-μm disc filter. Then, 15 ml of filtered lentivirus supernatant was concentrated through a 100-kDa ultrafiltration tube at 1500 × *g* and 4°C for 1 hr. Ph$^+$ leukemia cell lines were infected with concentrated lentivirus supernatant containing 8 μg/ml polybrene (H9268, Sigma). The culture plate or dish was centrifuged in a horizontal rotor centrifuge at 2000 × *g* and 32°C for 1.5 hr. After 48 hr, the viral particles were replaced with fresh medium, and 3 μg/ml puromycin was added for selection for another 48–72 hr. Protein expression levels were analyzed by immunoblotting with the antibodies of anti-RAPSYN (ab118491, Abcam) or SRC (11097-1-AP, Proteintech).

## Preparation of stable *RAPSN*-WT, *RAPSN*-Y336F, or *SRC* expression cell lines

The lentiviruses for overexpressing *RAPSN*-WT, *SRC* (both with corresponding empty vector OE-NC1), or *RAPSN*-Y336F (with corresponding empty vector OE-NC2) were obtained from GenePharma. The volume of virus required was calculated using the following equation: Ph$^+$ leukemia cell lines were infected using the spin-infection method described above. After infection for 48 hr, the viral particles were replaced with fresh medium. Stable RAPSYN-WT, RAPSYNY-336F, or SRC expressing cells were selected in the presence of 3 μg/ml puromycin for 48–72 hr. Protein expression levels were analyzed by immunoblotting with anti-RAPSYN or anti-SRC antibodies.

$$\text{virus volume} = \frac{\text{MOI} \times \text{cell number}}{\text{virus titer}}$$

## Protein expression and purification

Recombinant pGEX-4T-1-GST-RAPSYN plasmid were transformed into the ArcticExpress (DE3) pRARE2 competent *E. coli* cells (AYBIO-G6023, ANGYUBIO) and treated with 0.4 mM isopropyl-β-D-thiogalactoside (367-93-1, Sangon Biotech) to induce fusion protein expression at 18°C. After 50 hr, bacterial cells were harvested, resuspended in PBS (0.14 M NaCl, 2.68 mM KCl, 2 mM $KH_2PO_4$, 0.01 M $Na_2HPO_4$, 10 mM DTT, pH 7.4), and sonicated on ice. Precipitates were removed from cell lysates by centrifugation. Recombinant GST-RAPSYN was purified from the supernatant by GST-affinity chromatography (SA010010, Smart-Lifescience) and size-exclusion chromatography (17-0060-01, GE Healthcare). Purified GST-RAPSYN protein was digested with thrombin (T8021, Solarbio) for 6 hr at 4°C to remove GST tag. Recombinant pcDNA3.1-His-BCR-ABL plasmids were transfected into HEK293T cells and were collected after 48 hr culturing. The cells were then lysed on ice using NP-40 lysis buffer with a protease inhibitor cocktail. Cell lysates were centrifuged, and then the supernatant fraction was incubated with anti-BCR-ABL antibody overnight at 4°C and subsequently with protein A magnetic beads (73778, Cell Signaling Technology) for another 4 hr at 4°C. The bead complexes were washed three times with washing buffer (25 mM Tris–HCl, 0.15 M NaCl, 0.005% Tween-20, pH 7.5), eluted with

elution buffer (0.1 M glycine, pH 2.0), and mixed with neutralization buffer (1 M Tris–HCl, pH 9.0) for neutralization of purified protein.

## Protein stability assay

$RAPSN^{WT}$, $RAPSN^{Y336F}$, or BCR-ABL-transfected $Ph^+$ leukemia cells were incubated with 100 mg/ml cycloheximide (CHX, A8244; Cell Signaling Technology) for indicated time points. Cells were harvested and lysed on ice using NP-40 lysis buffer supplemented with a protease inhibitor cocktail. The supernatant was collected and subjected to immunoblotting using anti-RAPSYN or anti-SRC antibodies.

## Quantitative reverse transcription-PCR

High-quality RNA was isolated from cells or tissues using Trizol reagent (AJF1807A, Takara) according to the manufacturer's instructions. cDNA was synthesized from 1 µg of total RNA using HiScriptIIRT SuperMix for qPCR (R233-01, Vazyme). The ChamQ SYBR qPCR Master Mix (Q331-02, Vazyme) was used for two-step reverse transcription-PCR analysis on an Applied Biosystems StepOnePlus Real-Time PCR instrument. The samples were analyzed in triplicate. The expression value of target gene in a given sample was normalized to the corresponding expression of *ACTB* or *GAPDH*. The $2^{-\Delta\Delta Ct}$ method was used to calculate relative expression of target genes. The primers used are listed in *Supplementary file 1*.

## Cytotoxicity assay

Lentiviruses co-expressing GFP were used to assess the toxicity of shRNAs. Flow cytometry was performed 2 days after shRNA transduction to determine initial GFP-positive proportion of live cells for each shRNA. Subsequently, the cells were sampled every 2 days over the time. The GFP-positive proportion at each time point was normalized to that of day 2. Each shRNA experiment was performed at least three times for individual cell line.

## Statistical analysis

All in vitro experiments were repeated at least three times. Animals were randomly assigned to different groups for each in vivo study. Kaplan–Meier survival analysis was used for all survival studies, and the log-rank test was used to determine significant differences between groups. Differences with *p < 0.05, **p < 0.01, ***p < 0.001, and ****p < 0.0001 were considered significant. Prism 8 (GraphPad Software, CA, USA) was used for statistical analysis. Representative results from at least three independent replicates are shown. Data are presented as mean ± standard deviation (SD), and significant differences were determined using Student's *t*-test, unpaired Student's *t*-test, or one-way analysis of variance test.

## Acknowledgements

This work was supported by the grants of National Key R&D Program of China (2018YFA0902000), National Science Foundation of China (No. 81872924, 81973386, and 82002971), 'Double First-Class' University Project (CPU2022QZ014), Project Program of the State Key Laboratory of Natural Medicines, China Pharmaceutical University (SKLNMZZ202201), and PAPD of Jiangsu Province.

## Additional information

### Competing interests

Mengya Zhao, Beiying Dai, Xiaodong Li, Yixin Zhang, Shuzhen Wang, Yong Yang, Yijun Chen: listed as an inventor on Chinese patent 202210107464.7 (patent protection filed for by China Pharmaceutical University on RAPSYN). The other authors declare that no competing interests exist.

## Funding

| Funder | Grant reference number | Author |
| --- | --- | --- |
| National Key Research and Development Program of China | 2018YFA0902000 | Yijun Chen |
| National Natural Science Foundation of China | 81973386 | Yijun Chen |
| National Natural Science Foundation of China | 81872924 | Shuzhen Wang |
| National Natural Science Foundation of China | 82002971 | Beiying Dai |
| "Double First-Class" University Project | CPU2022QZ014 | Yijun Chen |
| Project Program of the State Key Laboratory of Natural Medicines | SKLNMZZ202201 | Yijun Chen |
| Key Research and Development Project of Guangdong Province | 2022B1111070004 | Yijun Chen |

The funders had no role in study design, data collection, and interpretation, or the decision to submit the work for publication.

## Author contributions

Mengya Zhao, Data curation, Formal analysis, Investigation, Visualization, Methodology, Writing – original draft; Beiying Dai, Data curation, Formal analysis, Funding acquisition, Investigation, Methodology, Writing – original draft, Project administration; Xiaodong Li, Data curation, Formal analysis, Investigation, Methodology; Yixin Zhang, Data curation, Formal analysis, Investigation, Visualization, Methodology; Chun Qiao, Yaru Qin, Zhao Li, Qingmei Li, Data curation; Shuzhen Wang, Conceptualization, Supervision, Funding acquisition, Project administration, Writing – review and editing; Yong Yang, Conceptualization, Resources, Supervision; Yijun Chen, Conceptualization, Resources, Data curation, Supervision, Funding acquisition, Writing – review and editing

## Author ORCIDs

Mengya Zhao https://orcid.org/0000-0002-6339-1892
Shuzhen Wang https://orcid.org/0000-0003-3869-2463
Yijun Chen https://orcid.org/0000-0002-4920-152X

## Ethics

This study was approved by the ethics committee of the First Affiliated Hospital of Nanjing Medical University (2019-SR-485.A1). Human peripheral blood samples were obtained from the remaining material utilized for routine laboratory tests at the First Affiliated Hospital of Nanjing Medical University (Nanjing, China) and derived from 21 patients with Ph+ chronic myeloid leukemia (CML), and six healthy volunteers. And one human bone marrow sample of Ph+ acute lymphoblastic leukemia patient was obtained from the remaining material utilized for routine laboratory tests at the First Affiliated Hospital of Nanjing Medical University (Nanjing, China).

All animal experiments were performed in accordance with the National Institutes of Health Guide for the Care and Use of Laboratory Animals with the approval of the Center for New Drug Evaluation and Research, China Pharmaceutical University (approval number: B20190925-1; Nanjing, China).

Reviewer #1 (Public Review): https://doi.org/10.7554/eLife.88375.3.sa1
Reviewer #2 (Public Review): https://doi.org/10.7554/eLife.88375.3.sa2
Author response https://doi.org/10.7554/eLife.88375.3.sa3

# Additional files

## Supplementary files
• Supplementary file 1. Sequences of shRNA and primer pairs used for real-time PCR to determine the mRNA levels.

• MDAR checklist

## Data availability
All data generated or analyzed during this study are included in the manuscript and supporting files.

The following previously published datasets were used:

| Author(s) | Year | Dataset title | Dataset URL | Database and Identifier |
|---|---|---|---|---|
| Kohlmann A, Kipps TJ, Rassenti LZ, Downing JR, Shurtleff SA, Mills KI, Gilkes AF, Hofmann WK, Basso G, Dell'orto MC, Foà R, Chiaretti S, De Vos J, Rauhut S, Papenhausen PR, Hernández JM, Lumbreras E, Yeoh AE, Koay ES, Li R, Haferlach T | 2009 | An international standardization programme towards the application of gene expression profiling in routine leukaemia diagnostics: the Microarray Innovations in LEukemia study prephase | https://www.ncbi.nlm.nih.gov/geo/query/acc.cgi?acc=GSE13159 | NCBI Gene Expression Omnibus, GSE13159 |
| Haferlach T, Kohlmann A, Wieczore KL, Basso G, Kronnie GT, Béné MC, De Vos J, Hernández JM, Hofmann WK, Mills KI, Gilkes A, Chiaretti S, Shurtleff SA, Kipps TJ, Rassenti LZ, Yeoh AE, Papenhausen PT, Liu WM, Williams PM, Foa R | 2009 | Clinical utility of microarray-based gene expression profiling in the diagnosis and subclassification of leukemia: report from the International Microarray Innovations in Leukemia Study Group | https://www.ncbi.nlm.nih.gov/geo/query/acc.cgi?acc=GSE13204 | NCBI Gene Expression Omnibus, GSE13204 |
| Herrmann H, Sadovnik I, Eisenwort G, Rülicke T, Blatt K, Herndlhofer S, Willmann M, Stefanz IG, Baumgartner S, Greiner G, Schulenburg A, Mueller N, Rabitsch W, Bilban M, Hoermann G, Streubel B, Vallera DA, Sperr WR, Valent P | 2020 | Delineation of target expression profiles in CD34+/CD38- and CD34+/CD38+ stem and progenitor cells in AML and CMLIG | https://www.ncbi.nlm.nih.gov/geo/query/acc.cgi?acc=GSE138883 | NCBI Gene Expression Omnibus, GSE138883 |
| Sinnakannu JR, Lee KL, Cheng S, Li J, Yu M, Tan SP, CCH Ong, Li H, Than H, Anczuków-Camarda O, Krainer AR, Roca X, Rozen SG, Iqbal J, Yang H, Chuah C, Ong ST | 2020 | SRSF1 mediates cytokine-induced impaired imatinib sensitivity in chronic myeloid leukemia | https://www.ncbi.nlm.nih.gov/geo/query/acc.cgi?acc=GSE140385 | NCBI Gene Expression Omnibus, GSE140385 |

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
