## [Editor Report · eLife assessment]

In this **important** study, the authors describe a novel function for RAPSYN in bcr-abl fusion associated leukemia, presenting **convincing** evidence that RAPSYN stabilizes the oncogenic BCR-ABL fusion protein. Compared to an earlier version of the manuscript, the authors have added data using primary samples that strengthen the conclusions.

---

## [Referee Report · Reviewer #1 (Public Review)]

The manuscript by Zhao et al describes the identification of RAPSYN, a NEDD8 E3 ligase previously studied for its role in acetylcholine receptor clustering and neuromuscular junction formation, as a factor promoting the stabilisation of the BCR-ABL oncogene in Chronic Myeloid Leukemia (CML) cells. The authors have identified that NEDDylation of BCR-ABL by RAPSYN antagonises its poly-ubiquitin and subsequent proteasome-based degradation. Knocking down RAPSYN with shRNA led to increased poly-ubiquitination and faster turnover of BCR-ABL. Furthermore, they describe that SRC-dependent phosphorylation of RAPSYN facilitates its NEDD8-ligase activity.

The authors' findings are primarily rooted in a series of well-conducted in vitro experiments using two CML cell lines, K562 and MEG-01. They have performed some further validations using primary CML samples, which have strengthened their claims.

The author's initial discoveries have come from interrogating a number of publicly available gene expression datasets, both microarray-based and RNA-seq, which revealed that RAPSYN is increased at the protein level but that RNA levels are not different between healthy and CML samples. This is a very interesting observation which warrants further future investigation.

The conclusions of this revised manuscript are broadly supported by the data and the analyses. It also describes novel findings that can spur future studies, both into the basic cellular biology of CML as well as into potential new therapeutic strategies.

Comments on revised version:

I thank the authors for addressing my concerns in the initial review. The revised manuscript with additional data is much stronger.

---

## [Referee Report · Reviewer #2 (Public Review)]

In this study the authors aim to elucidate the role of RAPSYN in BCR-ABL-mediated leukemogenesis. RAPSYN is mainly known as a scaffolding protein for anchoring acetylcholine receptors (AChRs) to the cytoskeleton in muscle cells, facilitating AChR clustering through neddylation (Li et al., 2016). The authors demonstrate, through a broad and rigorous array of biochemical assays, that RAPSYN also plays a crucial role in the neddylation of BCR-ABL in leukemia cells. Their results indicate that this process shields BCR-ABL from ubiquitination and subsequent degradation, likely through a mechanism involving competition for binding with the BCR-ABL ubiquitin ligase c-CBL. In addition, the authors delve into the regulatory mechanisms underlying RAPSYN stability, demonstrating that it is enhanced through phosphorylation by SRC. This discovery further deepens our understanding of the complex dynamics of the molecular interactions that regulate BCR-ABL stability in leukemia.

To confirm the physiological significance of their findings, the authors effectively utilize cell viability assays and in vivo models. The integration of these approaches lends strength and validity to their conclusions.

The implications of the findings presented in this study are important, particularly in relation to our understanding of the pathogenesis and potential therapeutic strategies for Philadelphia chromosome-positive leukemias. By illuminating the role of RAPSYN in the regulation of BCR-ABL stability, this research potentially uncovers avenues for the development of targeted therapies, making a significant contribution to the field.

Two areas of the study could benefit from additional validation and exploration:

(1) The authors propose that targeting RAPSYN in Ph+ leukemia could have a high therapeutic index, suggesting that inhibition of RAPSYN may lead to cytotoxicity in Ph+ leukemia with high specificity and minimal side effects. The authors now include data showing RAPSYN knockdown in HS-5 cells does not affect cell growth (Figure 1C), supporting this assertion. This observation presents a contrast to DepMap data (https://depmap.org/), where RNAi and CRISPR-mediated RAPSYN depletion across hundreds of cell lines does not exhibit obvious differential effects on cell viability compared to Ph+ leukemia cell lines. Therefore, while the current results are promising, they call for additional validation by future studies to confirm RAPSYN as a viable therapeutic target in this context.

(2) A particularly notable yet underexplored aspect of this study is the observed disparity between RAPSYN protein and mRNA levels in Ph+ patient samples and cell lines. There is a marked enrichment of RAPSYN protein levels (Figure 1A, B) despite seemingly unchanged mRNA levels (Supplementary Figure 1 A-C). The authors convincingly demonstrate that RAPSYN stabilizes BCR-ABL, while SRC-mediated phosphorylation in turn stabilizes RAPSYN. This points to a specific, SRC-driven stabilization mechanism of RAPSYN in the Ph+ leukemia context. Consequently, the question arises whether BCR-ABL (through activation of SRC) reciprocally stabilize RAPSYN? Exploring the effects of BCR-ABL depletion on RAPSYN levels could shed light on this potential two-way stabilization mechanism, offering deeper insight into the complex molecular dynamics of RAPSYN and BCR-ABL in Ph+ leukemias.

In conclusion, this study represents a pivotal advancement in our understanding of Philadelphia chromosome-positive leukemias. It uniquely positions RAPSYN, a protein previously not associated with leukemogenesis, as a key regulator of BCR-ABL stability. Future research is essential to establish RAPSYN's potential as a therapeutic target and to more comprehensively understand its role in this context.

Comments on revised version:

I acknowledge and appreciate the author responses. Below are our comments on each reply:

Reply 1: Your response and the inclusion of data regarding RAPSYN knockdown in HS-5 cells adequately address the concerns.

Reply 2: The issue of the disparity between RAPSYN protein and mRNA levels in Ph+ leukemias has not sufficiently been resolved. Refer to point 2 in the revised review for more details. If conducting the proposed experiment is not feasible, I recommend a more thorough discussion in the manuscript to address and hypothesize about the causes of this discrepancy between protein and mRNA levels.

Reply 3: Your rationale for not performing additional assays with inactive mutants is satisfactory.

Reply 4: The clarification provided in your revision of the method section and the reorganization of Figure 6 successfully resolve the previously noted discrepancies. However, to ensure consistency and clarity across the paper, I recommend that you also specify the batches of constructs/viruses used in other relevant figures, such as Figure 1E.

Reply 5: The clarification provided on the immunoblots sufficiently addresses the concern raised.

---

## [Author Response]

The following is the authors’ response to the original reviews.

**Reviewer #1 (Public Review):**
(1) The authors' findings are primarily rooted in a series of well-conducted in vitro experiments using two CML cell lines, K562 and MEG-01. While the findings are interesting and novel, further work to corroborate these findings in primary CML samples would have greatly strengthened the potential real-world relevance of these discoveries. The authors appear to have some PBMCs from primary CML patients and a BM sample from a Ph+ ALL in which they performed western blot analyses (Fig 1). Couldn't these samples have been used to at least confirm some of the key discoveries? For example, the neddylation of BCR-ABL, or; sensitivity of primary leukemic cells to RAPSYN knockdown, and/or; phosphorylation of RAPSYN by SRC?

We agree with your points and really appreciate your comments. To demonstrate the clinical relevance, we have conducted a series of experiments to address your concerns.

(1) after a thorough optimization on the transduction process, we have managed to show that shRNA-mediated gene silencing of RAPSYN impaired the growth of primary CML samples. These additional data are presented as Figure 1D in the revised manuscript with its corresponding figure legend and description, lines 136-141.

(2) we have invested tremendous time and effort to deal with “key discoveries” regardless of the almost impossible task with a great technical difficulty. With 5 mL (ethical approval) of PBMCs on hands, we have finally managed to confirm BCR-ABL neddylation by IP from two newly acquired CML patients. The results are as presented in Figure 2F in the revised manuscript with its corresponding figure legend and description, lines 186-187.

(2) The authors initially interrogated a fairly dated (circa 2009) microarray-based primary dataset to show that the increase in RAPSYN is primarily a post-transcriptional event, as mRNA levels are not different between healthy and CML samples. It would be interesting to see whether differences might be more readily seen in more recent RNA-seq datasets from CML patients, given the well-known differences in sensitivity between the two platforms. Additionally, I wonder if there would be transcriptional signatures of increased NEDDylation (or RAPSYN-induced NEDDylation) that could be interrogated in primary samples? Furthermore, there are proteomics datasets of CML cells made resistant to TKIs (through in vitro selection experiments) that could be interrogated for independent validation of the authors' discoveries. For example: from K562 cells, PMID: 30730747 or PMID: 34922009.

Thank you very much for your constructive comments. Based on your suggestion, we have (1) analyzed mRNA level of RAPSYN in RNA-seq datasets GSE13159 (2009), GSE138883 (2020) and GSE140385 (2020), indicating no difference between CML patients and healthy donors. We have included the results in Figure1-figure supplementary 1A and in the revised manuscript (lines 123-127); (2) examined the RNA levels of RAPSYN-related neddylation enzymes, including E1 (NAE1), E2 (UBE2M), NEDD8 and NEDP1 in these databases, and no significant differences of these neddylation-related genes were found between CML patients and healthy donors as well (Supplementary Figure 2C, lines 168-172).

We have also analyzed the proteomics datasets from PMID: 30730747 and PMID: 34922009 according to your suggestion. Unfortunately, no information on RAPSYN expression is available in these datasets. To avoid potential negligence, we have examined all CML-related proteomics datasets from 2002 to 2024, still resulting in no information about protein expression of RAPSYN. Consequently, our finding on the higher expression of RAPSYN in the PBMCs of Ph+ patients in this study appears to be an observation for the first time. And we believe that our results should be more clinically relevant than those, if any, from the cells by in vitro selection.

**Reviewer #2 (Public Review):**
Most of the conclusions drawn in this paper are well supported by data, but some aspects of the data need to be clarified and extended:(1) The authors propose that targeting RAPSYN in Ph+ leukemia could have a high therapeutic index, suggesting that inhibition of RAPSYN may lead to cytotoxicity in Ph+ leukemia with high specificity and minimal side effects. To substantiate this assertion, the authors should investigate the impact on cell viability upon RAPSYN knockdown in non-Ph leukemic cell lines or HS-5 cells (similar to Figure 1C), despite their lower RAPSYN protein levels.

We appreciate your valuable comments. When we used shRNA to knockdown the expression of RAPSYN in HS-5 cells, it did not affect the cell growth of HS-5 cells. We have included the data in Figure 1C, modified its figure legend, and added corresponding description, lines 136-141.

(2) The authors intriguingly show that the protein levels of RAPSYN are significantly enriched in Ph+ patient samples and cell lines (Figure 1A, B), even though the mRNA levels remain unchanged (Supplementary Figure 1 A-C). This observation merits a clear explanation in the context of the presented results. The data in the manuscript does imply a feedforward loop mechanism (Figure 7), where BCR-ABL activates SRC, which subsequently stabilizes RAPSYN, which in turn helps protect BCR-ABL from c-CBL-mediated degradation. If this is the working hypothesis, it would be beneficial for the reader to see supporting evidence.

Thank you very much for pointing out the issue. We have realized the inappropriateness of Figure 7, which was originally placed as a summarizing figure. To avoid potential confusion and misleading, this figure has been deleted, which does not affect the results and conclusions of this study. In addition, the differences on mRNA levels and protein expressions have been responded to Reviewer #1.

(3) The authors present compelling evidence to suggest that RAPSYN may possess direct NEDD8-ligase activity on BCR-ABL. To strengthen this claim, it may be valuable to conduct further assays involving a ligase-deficient mutant, such as C366A, beyond its use in Figure 2J. Incorporating this mutant into the in vitro assay illustrated in Figure 2K, for instance, could offer substantial validation for the claim. In addition, showing whether the ligase-deficient mutant is capable of phenocopying the phosphorylation-mutant Y336F, as showcased in Figures 5E, F, and 6D, F, would be beneficial.

We are grateful to your comments. In the manuscript, we have provided sufficient data to support the direct neddylation of BCR-ABL by RAPSYN, as you commented “The authors present compelling evidence to suggest that RAPSYN may possess direct NEDD8-ligase activity on BCR-ABL.”. Cys366 was previously demonstrated as the catalytic residue essential for E3 activity of RAPSYN (Li et al. 2016, PMID: 27839998), and the phosphorylation at Phe336 was thoroughly verified by site-directed mutagenesis and the treatments of SRC-specific inhibitor saracatinib in present cellular experiments. Therefore, while we fully respect your opinions, we do not think it would be necessary to perform tedious in vitro reactions for expected negative results, which was the reason for us not to conduct enzymatic reactions with known inactive mutants, such as C366A and Y336F, in the first place.

(4) The observations presented in Figures 6 C-G require additional clarification. Notably, there are discrepancies in relative cell viability effects in K562 cells, and to some extent in MEG-01 cells, under conditions that are indicated as being either identical or highly similar. For instance, this inconsistency is observable when comparing the left panels of Figure 6C and 6D in the case of NC overexpression + shSRC#2, and the left panels of Figure 6E and 6G with NC overexpression or shNC, respectively. Listing potential causes of these discrepancies would strengthen the overall validity of the findings and their subsequent interpretation.

Thank you for your comments and apologize for the confusion. To make a meaningful comparison, we have revised the method part “Preparation of stable RAPSYNWT, RAPSYNY336F or SRC expression cell lines” (lines 625-627) and reorganized Figure 6 to reflect the differences on the negative controls. In fact, we first used LV6 (EF-1a/Puro; OE-NC1) vector for the overexpression of RAPSYNWT and SRC. Due to low expression level with LV6 and long period of time for subsequent selection, we switched to LV18 (CMV/Puro; OE-NC2) for the overexpression of RAPSYNY336F. Since the sensitivities of K562/MEG01-OE-NC cells to shSRC transduction in Figure 6C (now revised to K562/MEG01-OE-NC1) and 6D (now revised to K562/MEG01-OE-NC2) were noticeably different, we have separated RAPSYNWT and RAPSYNY336F cells as 6C and 6D with their own corresponding empty vector as negative control, instead of merging the results into a single figure with one negative control of OE-NC. In addition, given the fact that K562/MEG01 cells reacted differently upon saracatinib treatments after transduction with the empty vector, we have also distinguished the negative controls as OE-NC1 in Figure 6E, OE-NC2 in Figure 6F and shNC in Figure 6G. Afterall, the transduction of K562/MEG01 cells with different expression vectors and viral particles caused the discrepancies in the experiments of cell viability, which has been clarified by reorganizing Figure 6 in the revision.

(5) Throughout the manuscript, immunoblots which showcase immunoprecipitations of BCR-ABL or His-BCR-ABL depict poly-neddylation (e.g. Figures 2E-M, 3D-G, and 5A-E) and poly-ubiquitination (e.g. Figures 3D-G) patterns/smears where these patterns seem to extend below the molecular weight of BCR-ABL. To enhance clarity, it would be valuable for the authors to provide an explanation in the text or the figure legend for this observation. Is it reflective of potential degradation of BCR-ABL or is there another explanation behind it?

Thank you for your valuable comments. After carefully checking original immunoblots, we have ascertained that the protein band of BCR-ABL was at 250 KDa and the smear bands appeared to be higher than 250 KDa were likely caused by the conjugation of NEDD8 (neddylation) or Ubiquitin (ubiquitination) onto BCR-ABL. Regarding the molecular weight of modified BCR-ABL lower than expected, whether it is a common feature as previously reported (Mao, J., et al, 2010, PMID: 21118980) or possible degradation during the modification process or sample preparation requires further investigation. We have corrected the labeling of figures in the revised manuscript.

**Reviewer #1 (Recommendations For The Authors):**
(1) It would really nail the real-world relevance of these nice findings if the authors are able to confirm some aspects of their cell line-based discoveries in publicly available 'omics datasets generated from primary CML samples. I have suggested some of these in the public review as well.Alternatively, if they are able to investigate samples from murine CML models (eg. BALB/c CML models), it would represent a step towards real-world relevance.

Thank you very much for your constructive comments. According to your suggestion, we have examined and analyzed RAPSYN mRNA and protein in updated and publicly available datasets as replied in the public response.

(2) The Discussion repeats some of the information already presented in the Introduction (for example, lines 311-327 of the merged document, or lines 349-358). I would urge the authors to instead expand more about how RAPSYN might be upregulated at the post-transcriptional level, or its potential post-translational regulation by SRC-mediated phosphorylation.

Thanks for your constructive suggestion. We have re-written this part according to your suggestion and marked in red color in the revised manuscript, lines 319-325 and lines 351-378.

(3) There are instances of clunky phrases/grammatical mistakes in the manuscript which detract from its readability (eg: lines 142-143: "...empty body transduced shRAPSN#3 or K562 cells into...."; lines 163-164: "Despite AChR subunits α7, M2, M3, and M4 were expressed in all tested cells, no change..."; line 178: "Preeminent BCR-ABL neddylation was detected in..."). A closer proof-reading of the final manuscript is advisable.

We appreciate the valuable comments. We have made changes for improvement, which is marked in red color in the revised manuscript, lines 145-147, lines 166-168 and line 185.

(4) The western blot in Fig 5C (particularly the control "OE-NC" of K562) looks drastically different from the corresponding control lanes in Figs 5A and 5B.Similarly, the cell viability curves presented in Fig 6D and 6F (for both K562 and MEG-01, control conditions) look very different from the corresponding curves in Figs 6A and 6B.

We appreciate for your valuable comments. Because we accidently used the imagines with different exposure time, the western blots in Fig 5C (particularly the control "OE-NC" of K562) look very different from corresponding control lanes in Figs 5A and 5B. We have replaced images with the same exposure time in the revised manuscript.

For readers to clearly understand, we have revised the method part “Preparation of stable RAPSYNWT, RAPSYNY336F or SRC expression cell lines” (lines 625-627) and related figure legends to reflect the differences.

We have publicly responded the discrepancy on cell viability.

**Reviewer #2 (Recommendations For The Authors):**
In reviewing your study, I must insist that the completeness and robustness of your work would significantly benefit from a more exhaustive listing of the antibodies used for immunoblotting and immunoprecipitation within the Materials and Methods section. A number of antibodies have been accounted for, however, crucial ones targeting BCR-ABL, c-CBL, Ubiquitin, NEDD8, HA, Myc, and others appear to be omitted. To maintain rigorous scientific standards, I strongly encourage you to include these.

We appreciate your comments. We have carefully checked the section of Methods and added detailed information of antibodies for Immunoblotting and Immunoprecipitation in the revised manuscript, lines 502-516.